# In-Flight Estimation of Instrument Spectral Response Functions Using Sparse Representations

Jihanne El Haouari[1,2], Jean-Michel Gaucel[3], Christelle Pittet[4], Jean-Yves Tourneret[1,2], and Herwig Wendt[2,5]

[1]TéSA laboratory, Toulouse, France
[2]IRIT, Univ. Toulouse, Toulouse, France
[3]Thales Alenia Space Cannes, France
[4]Centre National d'Etudes Spatiales (CNES), Centre Spatial de Toulouse, France
[5]Centre National de Recherche Scientifique (CNRS)

**Abstract.** High resolution spectrometers are composed of different optical elements and detectors that must be modeled as accurately as possible. Specifically, accurate estimates of Instrument Spectral Response Functions (ISRFs) are critical in order not to compromise the retrieval of trace gas concentrations from spectral measurements. Currently, parametric models are used to estimate these response functions. However, these models cannot always take into account the diversity of ISRF shapes that are encountered in practical applications. This paper studies a new ISRF estimation method based on a sparse representation of the ISRF in a dictionary. The proposed method is shown to be very competitive when compared to parametric models, yielding up to one order of magnitude smaller normalized ISRF estimation errors. The method is applied to different high-resolution spectrometers, demonstrating its reproducibility for multiple remote sensing missions.

## 1 Introduction

Space remote sensing makes it possible to remotely measure the composition of the atmosphere or the troposphere and to retrieve trace gas concentrations. It can also be used to monitor molecule fluxes at the Earth's surface, as is the case for the MicroCarb mission that is designed to monitor $CO_2$ fluxes (Cansot et al., 2022) in order to provide a better understanding of the carbon cycle, which is important in the context of climate change. This can be done by analyzing the interaction of the atmosphere with natural radiation, such as the sunlight, or artificial radiation, generated for example by a laser. Indeed, the presence of some molecules in the path of radiation modifies its spectral content at the characteristic wavelengths of the different elements. The information directly obtained from satellites is the atmospheric spectrum. By considering some specific wavelengths of interest, it is possible to determine the concentration of the desired trace gases in a column of atmosphere by comparing these measured spectra with a reference spectrum obtained using a radiative transfer model.

The instruments used for gas concentration estimation are high resolution spectrometers. Spectrometers consist mainly of an optical part (for example composed of a slit, a telescope and dispersive grating) and a detector. In this configuration, the telescope projects the image of the Earth onto the spectrometer slit and then onto the detector. Each pixel of the detector is associated with a spatial direction (called ACT for ACross Track) and a specific wavelength. A binning and an averaging along

the ACT axis are performed in order to improve the Signal to Noise Ratio (SNR). For each of the two parts (optical part and detector), a response function is defined, which leads to a continuous optical function and another function associated with each pixel of the detector. This results in a global response function associated with each pixel along the spectral axis, known as the Instrument Spectral Response Function (ISRF), associated with a specific wavelength. The ISRFs can vary significantly depending on the instrument considered and their shapes depend on the central wavelength, among other factors. The estimation of trace gas concentrations is an inversion process that is performed on the ground from spectrometer measurements and the instrument ISRFs. The accuracy of this estimation highly depends on the knowledge of these ISRFs for all pixels. For some missions, ISRFs are expected to be known with a normalized error less than $1\%$, which represents a significant challenge given that the variations in ISRF shape across the entire band frequently exceed this threshold.

Spectrometers are first calibrated on ground where their associated ISRFs are estimated experimentally. However, the ISRFs are subject to in-flight changes due to mechanical movements associated with the launch of the instruments, thermal changes in orbit, or certain sensitivities linked to the instrument itself (such as the MicroCarb's sensitivity to the scene). As a consequence, these ISRFs need to be re-estimated regularly in-flight throughout the mission. The principle of the estimation is to take a measurement of a spectrally known scene and to compare it with a spectral model of the scene convolved with the ISRFs at different wavelengths. Parametric models have been widely used in the literature to estimate ISRFs. Gaussian and generalized Gaussian parametric models (referred to as "Gauss" and "Super-Gauss") were proposed in (Beirle et al., 2017). Parametric models are attractive for their simplicity and small number of parameters. However, they are not flexible enough to represent the diversity of ISRF shapes adequately. The ISRF estimation problem and the most important parametric models that have been considered in the literature are detailed in Section 2.

The objective of this work is to overcome the limitations of the existing parametric ISRF estimation methods caused by their insufficient accuracy. To that end, we propose as a first major contribution a new estimation strategy based on sparse representations of the ISRFs in a dictionary of well chosen atoms. More precisely, the ISRFs are decomposed in a dictionary that is constructed using several ISRFs that are available from ground characterization for each instrument. The dictionary can also be updated iteratively on-line. For each instrument, each ISRF is then approximated by a linear combination of a small number of atoms of the dictionary associated with the instrument. The proposed approach is detailed in Section 3. We investigate and compare two different methods for obtaining the sparse representations of ISRFs.

As a second contribution, we conduct an extensive numerical study of the proposed ISRF estimation approach and compare it to parametric methods for datasets from several different spectrometers used in space missions, whose characteristics are detailed in Section 4. The main focus is on the MicroCarb instrument (Cansot et al., 2022), which is dedicated to study the atmospheric carbon dioxide and oxygen, with the objective of determining their concentrations at the Earth's surface. Additional results showing the applicability of the proposed methodology to other spectrometers are reported for the OCO-2 spectrometer (Lee et al., 2017), and complemented by results for several other spectrometers that are reported in the Supplementary Material (El Haouari et al., 2024).

Numerical results are reported in Section 5 and lead to conclude that the proposed method yields significantly improved flexibility and accuracy for ISRF estimation when compared to previous state-of-the-art parametric methods, consistently

through the different datasets and scenarios, with a small number of parameters that can be easily and efficiently estimated in real-time. Moreover, the method is shown to be robust with respect to design choices, to the noise corrupting the observed measurements, to ISRF changes depending on the scene or to possible mismatches on the prior knowledge on the ISRFs or reference spectra.

## 2 Existing models and estimation methods

### 2.1 ISRF estimation model

The ISRF, that is sometimes referred to as Instrument Line Shape (ILS) (Sun et al., 2017b) or Slit Function (Sun et al., 2017a), is a function that describes the response of an instrument to a given wavelength. In this work, we only consider the spectral information and thus each "pixel" $l$ is associated with a specific wavelength $\lambda_l$ yielding an ISRF at this wavelength.[1] The in-flight identification of ISRFs is obtained from scenes that are assumed to be perfectly known radiometrically and spectrally (such as Sun, Moon, uniform scenes such as desert, etc.), which are referred to as reference spectra. The principle of ISRF estimation is to determine the in-flight ISRFs for each wavelength $\lambda_l$ that minimize some similarity measure between the measured spectrum $s(\lambda_l)$ and the reference spectrum $r(\lambda)$ convolved with the ISRF denoted as $I_l(\lambda_l)$:

$$s(\lambda_l) = (r * I_l)(\lambda_l) = \int_{\mathbb{R}} r(\lambda_l - u)I_l(u)du, \quad l = 1, ..., N_\lambda, \tag{1}$$

where $*$ denotes convolution and $N_\lambda$ is the number of central wavelengths $\lambda_l$, each associated with one ISRF $I_l$. For practical purposes, this equation can be discretized leading to:

$$s(\lambda_l) \approx \sum_{n=-N/2}^{N/2} r(\lambda_l - n\Delta)I_l(n\Delta), \quad l = 1, ..., N_\lambda, \tag{2}$$

where $\Delta$ is the sampling period between two consecutive points of the ISRF, which is assumed to be regularly sampled. In other words, a vector $\boldsymbol{I}_l = [I_l(-\frac{N}{2}\Delta), ..., I_l(\frac{N}{2}\Delta)]^T \in \mathbb{R}^{N+1}$ needs to be estimated for each ISRF, corresponding to the values that it takes on the wavelength grid at which the ISRFs are sampled. $\boldsymbol{\Delta} = \{-\frac{N}{2}\Delta, ..., \frac{N}{2}\Delta\} \in \mathbb{R}^{N+1}$. The objective of the ISRF estimation problem is to solve the inverse problem (2) assuming knowledge of both the reference spectrum $r(\lambda)$ and the measurements $s(\lambda_l)$.

A major difficulty with the inverse problem (2) is that there is only one measurement per fixed wavelength $\lambda_l$, which makes it impossible to estimate the vector $\boldsymbol{I}_l$ without further assumptions. Two approaches can be used to make this estimation problem identifiable.[2] The first idea is to consider knowledge of several reference spectra for every wavelength. The problem

---

[1]In practice, the wavelength associated with the pixel is obtained as the center (maximum, median, or barycenter) of the measured ISRF at the given pixel. However, there are some effects, such as the smile (in ACT) or some gaps in our knowledge about the wavelengths (in along track), that can result in spectral shifts, which can degrade the estimation of ISRFs. These aspects are not considered in the present work. Thus, it is assumed that each pixel is associated with one wavelength which is known, and address the ISRF estimation problem by solving an inverse problem.

[2]Additional measurements could in principle be obtained experimentally using, e.g., a spectrally tunable on-board calibration source, albeit at extra cost.

is that this would not only require a sufficient number of calibration scenes to be available, but also that they substantially differ for each wavelength in order to provide complementary information on the shapes of the ISRFs. The second method, which is considered in this paper, has the advantage that it makes use of only one reference spectrum and is based on the assumption that the ISRFs for adjacent wavelengths $\lambda_l$ are similar, i.e., they exhibit slight variations along the spectral axis between $\lambda_l$ and $\lambda_{l+1}$. It is expected that the average of the normalized absolute error between the ISRFs in a window of $N_{\text{obs}} + 1$ observations and the central ISRF at wavelength $\lambda_l$ is below a given criterion for the ISRF estimation error. Note that the larger this variation, the more important the discrepancies in ISRF shapes. The small variation assumption is not valid for the whole set of wavelengths and the size of the sliding window must be adjusted in order to solve the ISRF estimation problem. This is a reasonable assumption for the ISRFs of real-world spectrometers. To estimate the ISRF at wavelength $\lambda_l$, we propose to consider a vector $\boldsymbol{s}_l = [s(\lambda_{l-\frac{N_{\text{obs}}}{2}}), ..., s(\lambda_{l+\frac{N_{\text{obs}}}{2}})]^T \in \mathbb{R}^{N_{\text{obs}}+1}$ of $N_{\text{obs}} + 1$ observations, including also those from the neighboring ISRFs. Rewritten in matrix form, (2) simplifies to:

$$\boldsymbol{s}_l = \boldsymbol{R}_l \boldsymbol{I}_l,$$

where $\boldsymbol{R}_l = [\boldsymbol{r}_{l-\frac{N_{\text{obs}}}{2}}, ..., \boldsymbol{r}_{l+\frac{N_{\text{obs}}}{2}}]^T \in \mathbb{R}^{(N_{\text{obs}}+1)\times(N+1)}$ contains the values $\boldsymbol{r}_l = [r(\lambda_l - \frac{N}{2}\Delta), ..., r(\lambda_l + \frac{N}{2}\Delta)] \in \mathbb{R}^{N+1}$ of the reference spectrum covered by the different ISRFs in the neighborhood (see algorithm in Appendix A1). Given a model for the ISRF, estimating $\boldsymbol{I}_l$ can then be conducted for each wavelength $\lambda_l$ by minimizing the residual error $||\boldsymbol{s}_l - \boldsymbol{R}_l \boldsymbol{I}_l||_2^2$.

## 2.2   Parametric models

It is difficult to analytically construct accurate forward models with a small number of parameters for ISRFs because they would
need to incorporate a significant number of "contributors" associated with the instrument optics (slit, mirror, lens, separator, dispersing element), the detector or the acquisition mode. The state of the art therefore considers simple parametric models. A classical way to model and estimate the ISRF at wavelength $\lambda_l$ is to use a parametric Gaussian model defined by:

$$\boldsymbol{I}_{l,\boldsymbol{\beta}_{\text{G}}}(x) = A_G \exp\left[-\frac{(\lambda_l - x - \mu_{\text{G}})^2}{2\sigma_G^2}\right], \quad l = 1, ..., N_\lambda, \quad x \in \boldsymbol{\Delta}, \tag{3}$$

where $\boldsymbol{\beta}_G = [A_{\text{G}}, \mu_{\text{G}}, \sigma_G^2]^T$ is the unknown vector of parameters to be estimated.

An alternative ISRF model was studied (Beirle et al., 2017) using a generalized Gaussian distribution referred to as "super-Gaussian" in order to better fit the ISRF shapes:

$$\boldsymbol{I}_{l,\boldsymbol{\beta}_{\text{SG}}}(x) = A_{\text{SG}} \exp\left[-\left|\frac{\lambda_l - x - \mu_{\text{SG}}}{w_{\text{SG}}}\right|^{k_{\text{SG}}}\right], \quad l = 1, ..., N_\lambda,, \quad x \in \boldsymbol{\Delta}, \tag{4}$$

where $\boldsymbol{\beta}_{\text{SG}} = [A_{\text{SG}}, \mu_{\text{SG}}, w_{\text{SG}}, k_{\text{SG}}]^T$ is the unknown parameter vector to estimate. This model reduces to the Gaussian model when $w_{\text{SG}} = 2\sigma_{\text{G}}^2$ and $k_{\text{SG}} = 2$. The parameters $w_{\text{SG}}$ and $k_{\text{SG}}$ are the scale and shape parameters of the distribution, allowing
more or less flat shapes to be modeled.

When using the parametric models (3) and (4), the ISRF estimation problem consists of estimating the unknown model parameters for each sliding window. This estimation can be performed using the least squares method, which minimizes the

following cost function:

$$C_l(\boldsymbol{\beta}) = \sum_{n=1}^{N+1} ||\boldsymbol{s}_l - \boldsymbol{R}_l \boldsymbol{I}_{l,\boldsymbol{\beta}}||_2^2, \quad l = 1, ..., N_\lambda, \tag{5}$$

where $\boldsymbol{\beta} \in \{\boldsymbol{\beta}_\mathrm{G}, \boldsymbol{\beta}_\mathrm{SG}\}$ is the unknown parameter vector and $\boldsymbol{I}_{l,\boldsymbol{\beta}} = [\boldsymbol{I}_{l,\boldsymbol{\beta}}(\delta_1), ..., \boldsymbol{I}_{l,\boldsymbol{\beta}}(\delta_{N+1})]^T$.

Simple parametric models, such as Gaussian or generalized Gaussian models, are attractive for their simplicity and small number of parameters, yet can struggle to take into account the variety of different ISRF shapes that can be observed in practice. An illustration is provided in Fig. 1, which shows examples of ISRFs for the MicroCarb mission. Clearly, these ISRFs cannot be accurately modeled by bell-shaped Gaussian distributions or by generalized Gaussians (because of the dip at the center, for example). This motivates the study of a new estimation method for ISRFs.

## 3 Sparse approximations of ISRFs

This paper investigates the use of sparse representations for ISRFs in a dictionary of well chosen atoms. Models based on sparse approximations and on dictionary learning have been widely and successfully used for different signal and image processing applications (Zhang et al., 2015). These applications include image denoising, image classification, image reconstruction, compressed sensing or dimensionality reduction and involve large varieties of signals and images (Figueiredo et al., 2007; Tošić and Frossard, 2011). However, sparse representations have never been investigated for ISRF estimation, which is precisely the objective of this work.

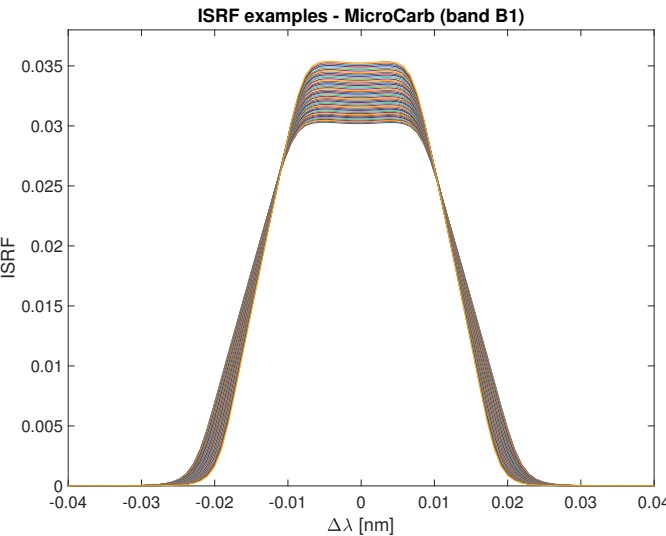

**Figure 1.** Illustration of a superposition of 1024 ISRFs with centered wavelengths $\lambda_l = 758.3, ..., 768.3$nm around their central wavelengths. The ISRFs have been simulated for the band B1 of the MicroCarb instrument using uniform scenes.

## 3.1 Construction of the dictionary

Sparse representations express a given signal as a linear combination of a small number of signals that belong to a collection of reference patterns, or atoms, which is called a dictionary. This paper proposes to decompose the ISRF in a dictionary of atoms $\boldsymbol{\Phi} \in \mathbb{R}^{(N+1) \times N_{\mathrm{D}}}$:

$$\boldsymbol{I}_l \approx \boldsymbol{I}_l^K = \boldsymbol{\Phi} \boldsymbol{\alpha}_l = \sum_{k=1}^{K} \boldsymbol{\Phi}_{\gamma_k} \alpha_{l,k}, \quad l = 1, ..., N_\lambda, \tag{6}$$

where $\boldsymbol{\Phi}_{\gamma_k}$ is the $\gamma_k$th selected atom, i.e., the $\gamma_k$th column of the dictionary $\boldsymbol{\Phi}$ and $\alpha_{l,k}$ is the corresponding non zero coefficient of the sparse vector $\boldsymbol{\alpha}_l = [\alpha_{l,1}, ..., \alpha_{l,K}]^T \in \mathbb{R}^{N_{\mathrm{D}}}$. The dictionary is built in such a way that linear combinations of a small number of its atoms (i.e., its columns) provide an efficient representation of the ISRF. Different methods allowing the dictionary to be built have been proposed in the literature. These methods are based on probabilistic learning, clustering, vector quantization or Bayesian inference (Tošić and Frossard, 2011). Dictionary learning usually involves a two-stage optimization structure, consisting first of a sparse coding step, to find the sparse vector $\boldsymbol{\alpha}_l$ which minimizes the objective function $||\boldsymbol{I}_l - \boldsymbol{\Phi} \boldsymbol{\alpha}_l||_2^2$ for a fixed dictionary $\boldsymbol{\Phi}$ and then a dictionary update step, where the dictionary is estimated given a fixed sparse vector $\boldsymbol{\alpha}_l$. Depending on the application, the dictionary can be updated using a closed form solution, gradient descent, or using ground truth data. In this work we investigate two different ways of building the dictionary $\boldsymbol{\Phi}$. The first method constructs $\boldsymbol{\Phi}$ by using the $N_{\mathrm{D}}$ singular vectors associated with the largest singular values of the SVD of a matrix composed of representatives ISRF examples, as described in the algorithm of Appendix A2. The second method uses the K-SVD algorithm of (Aharon et al., 2006), which belongs to the state of the art and is recalled in the algorithm of Appendix A5. The K-SVD algorithm is a generalization of the K-means algorithm in which the dictionary is updated by changing its columns separately and sequentially and applying $K$ singular value decompositions (SVDs) on an appropriate error matrix. Fig. 2 displays the first atoms of dictionaries constructed using these two methods for the band B1 of MicroCarb. These dictionaries are found to be similar, especially the two first atoms that correspond to the most energetic singular values. The two first atoms can be interpreted as the approximate average of all ISRFs used to build the dictionary (first atom), and a correction for adjusting the different widths of the ISRFs for different wavelengths (second atom), as seen in Fig. 1. The higher order atoms obtained with SVD and K-SVD are slightly different but with similar shapes overall.

## 3.2 Inverse problem

Assuming that the ISRF can decomposed in the dictionary $\boldsymbol{\Phi}$ as in (6), the measured spectrum can be written as follows:

$$\boldsymbol{s}_l \approx \boldsymbol{R}_l \boldsymbol{I}_l \approx \boldsymbol{R}_l \boldsymbol{\Phi} \boldsymbol{\alpha}_l = \boldsymbol{\Psi}_l \boldsymbol{\alpha}_l, \quad l = 1, ..., N_\lambda.$$

Thus, the ISRF estimation problem reduces to finding the sparse vector $\boldsymbol{\alpha}_l$ that minimizes the residual $||\boldsymbol{s}_l - \boldsymbol{\Psi}_l \boldsymbol{\alpha}_l||_2^2$. This sparse coding problem has been mathematically formulated in different ways (Zhang et al., 2015). One can use the $l_0$ pseudo-norm regularization $|| \cdot ||_0$ with a penalty parameter $\mu$, leading to the following problem:

$$\arg\min_{\boldsymbol{\alpha}_l} L(\boldsymbol{\alpha}_l, \mu) = \arg\min_{\boldsymbol{\alpha}_l} ||\boldsymbol{s}_l - \boldsymbol{\Psi}_l \boldsymbol{\alpha}_l||_2^2 + \mu ||\boldsymbol{\alpha}_l||_0, \quad l = 1, ..., N_\lambda. \tag{7}$$

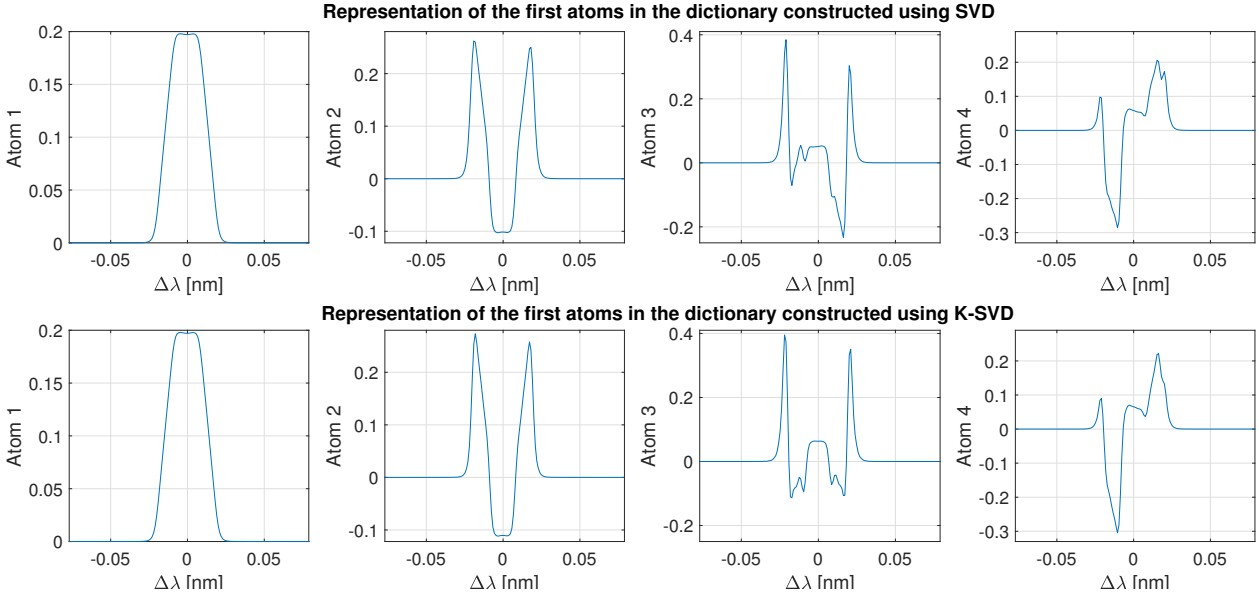

**Figure 2.** Representation of the four first atoms of the dictionary of ISRFs $\boldsymbol{\Phi}$ constructed using an SVD on the matrix of representative ISRFs (top) or using the K-SVD algorithm using the same matrix of representative ISRFs (bottom) for the MicroCarb spectrometer (band B1).

This problem is non-convex and NP-hard, and many approximations and heuristics have been proposed in the literature to find

an approximate solution. A standard method consists of using greedy algorithms such as the Orthogonal Matching Pursuit (OMP). OMP is a modification of the Matching Pursuit (MP) algorithm, which improves convergence by adding an orthogonalization step (Mallat and Zhang, 1993; Pati et al., 1993). The atoms of the dictionary that minimize the data fidelity term $||\boldsymbol{s}_l - \boldsymbol{\Psi}_l \boldsymbol{\alpha}_l||_2^2$ are iteratively determined by minimizing the remaining residual error. The OMP algorithm is summarized in the algorithm of Appendix A3. Another method replaces the pseudo-norm $l_0$ in (7) by the $l_1$ norm, which leads to a convex

problem known as the LASSO problem (Tan et al., 2015):

$$\arg\min_{\boldsymbol{\alpha}_l} L(\boldsymbol{\alpha}_l, \mu) = \arg\min_{\boldsymbol{\alpha}_l} ||\boldsymbol{s}_l - \boldsymbol{\Psi}_l \boldsymbol{\alpha}_l||_2^2 + \gamma ||\boldsymbol{\alpha}_l||_1, \quad l = 1, ..., N_\lambda, \tag{8}$$

and the related algorithms studied in, e.g., (Figueiredo et al., 2007; Kim et al., 2007).

The OMP and LASSO algorithms provide a highly flexible decomposition of the ISRF, as the choice of the dictionary is not constrained to a specific form. Indeed, the basis functions can be learned, for example by using the K-SVD algorithm in

conjunction with various Matching Pursuit algorithms. Another advantage of these methods is that they do not necessitate any prior assumption on the shape of the ISRFs (such as Gaussian ISRFs) and estimate them in a non-parametric way.

In the following, this paper compares the use of fixed dictionaries obtained by a single SVD, and dictionaries estimated by K-SVD (alternation between SVD to update the dictionary and OMP to update the sparse code). The proposed approach using OMP or LASSO (or other sparse formulations) and either fixed or re-estimated dictionaries will be referred to as SPIRIT for

"SParse representation of Instrument spectral Response functions using a dIcTionary".

## 4 Instruments, datasets & preprocessing

The spectrometers used in this study are passive pushbroom spectrometers, mainly hyperspectral dispersive spectrometers, such as the MicroCarb high-resolution spectrometer and the OCO-2 instrument. [3]

### 4.1 Synthetic data generation

Reference spectra used in this study were generated using the 4A/OP (Automatized Atmospheric Absorption Atlas) software (NOVELTIS et al., 2012). This software is based on a fast and accurate line-by-line transfer model that can be integrated in operational processing chains including inverse problem processing (Armante et al., 2013). It was selected as the official radiative model and reference code by CNES for the MicroCarb mission. The profiles originate from the Thermodynamical Initial Guess Retrieval (TIGR) database, which is hosted by Aeris data [4]. An example of profile was selected from this database

for the generation of a reference spectrum. The measured spectra were then obtained by convolving the reference spectrum with the ISRFs (normalized to area 1 for each instrument, see details in the next paragraphs) and embedded in additive Gaussian noise to generate representative measurements. The advantage of this data generation method is to provide ground truth ISRFs, which can be used to assess the performance of the different methods in a controlled scenario.

### 4.2 MicroCarb mission

MicroCarb is a mission developed by the Centre National des Études Spatiales (CNES) whose aim is to ensure continuity with other carbon measuring missions such as OCO-2 and GoSat, in order to monitor $CO_2$ fluxes at the Earth surface and determine $CO_2$ atmospheric concentrations. The MicroCarb mission uses a compact and low cost space instrument that will be smaller than the current spectrometers. The instrument is capable of acquiring four spectral bands with a single detector. The first band B1 (758.3-768.3) nm is an $O_2$ band with a spectral resolution of about 0.01 nm. The bands B2 (1596.7-1618.9 nm)

and B3(2023-2051 nm) with respective spectral resolutions of about 0.02 nm and 0.03 nm are sensitive to the concentration of $CO_2$ and have $CO_2$ absorption lines. The last band B4 (1264-1282.2 nm) is a second $O_2$ band with spectral resolution of about 0.02 nm. The wavelengths associated with this last band are closer to the $CO_2$ wavelength and can be used for validation of space-based greenhouse gas observation (Bertaux et al., 2020). The whole dataset has been delivered by the French Space Agency (CNES, Toulouse) containing 1024 ISRFs associated with 1024 spectral measurements for the different bands. The

data used for this experiment is the first band of MicroCarb with $N_\lambda = 1024$ ISRFs and a sample size $N = 895$. The design of the MicroCarb instrument, obtained from (Castelnau et al., 2019), is displayed in Fig. 3. More details about MicroCarb can

---

[3]Alternative designs, such as Fourier Transform InfraRed spectroscopy (FTIR) spectrometer are also employed in practice and the associated ISRFs can be obtained through the inverse Fourier transform. However, in certain applications, applying the Fourier transform can become more challenging (i.e., when undersampling is necessary or when the Optical Path Difference varies depending on the position). If the problem can be modeled as a linear inverse problem, sparse representation-based methods can be used with these spectrometers to estimate ISRFs. The proposed method is not specific to any instrument and can be applied to any instrument for which the problem can be formulated as a linear inverse problem.

[4]Data available at https://www.aerisdata.fr/en/projects/thermodynamical-initial-guess-retrieval-tigr/

be found on the CNES website. [5] A particularity of this mission is that the shapes of the ISRFs are strongly dependent on the scene observed by the instrument, which will be discussed in Sect. 5.3.3.

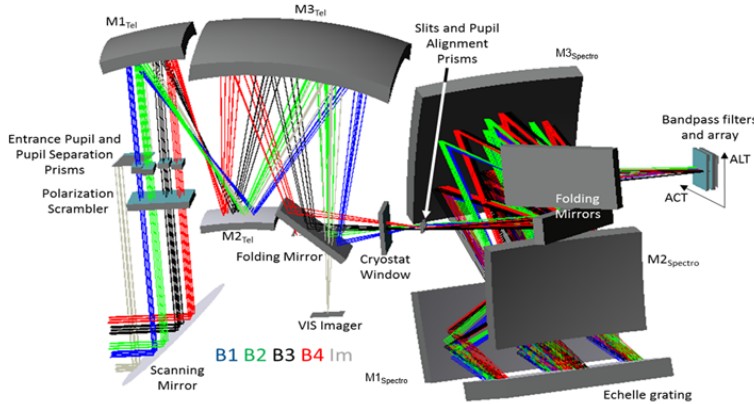

**Figure 3.** Principle design of the MicroCarb instrument reproduced from (Castelnau et al., 2019).

## 5 Results and discussion

### 5.1 Numerical experiments and performance evaluation

The performance of the different ISRF estimation methods is evaluated in terms of ISRF estimation quality and residual between the spectral measurements and their estimates. The quality of ISRF estimation can be quantified by the normalized absolute error between the ISRF and its estimate:

$$\mathrm{E}_l = \sum_{n=-N/2}^{N/2} |I_l(n\Delta) - \hat{I}_l(n\Delta)|.$$

Note that for the instruments studied here, the ISRFs are assumed to be normalized to unit area. The residual between the spectral measurements and their estimates is defined for each $\lambda_l$ by:

$$\rho_l = ||\boldsymbol{s}_l - \boldsymbol{r}_l \hat{\boldsymbol{I}}_l||_2^2,$$

and summarized for an entire band in terms of the average residual:

$$\rho = \frac{1}{N_\lambda} \sum_{l=1}^{N_\lambda} \rho_l.$$

In the MicroCarb mission, the ISRFs are considered to be well estimated when their normalized errors satisfy $\mathrm{E}_l < 1\%$ for each wavelength. The performance of 1% on the ISRF knowledge is an objective of the MicroCarb mission in order to provide an

---

[5]see CNES website: https://microcarb.cnes.fr/en

accurate determination of $CO_2$ concentrations. The 1% requirement accounts for uncertainty, acquisition noise of ISRFs and interpolation, and is used as a target in this work. The proposed SPIRIT method is compared to the parametric methods based on Gaussian and Super-Gaussian models. The parameters of these models are estimated using the non-linear least squares algorithm based on the Nelder-Mead optimization algorithm (Lagarias et al., 1998) (MATLAB function *fminsearch*). This iterative algorithm requires an initialisation and a stopping criterion. For the initialization of the Gaussian model, the mean $\mu_{G_0}$ was set to the sample mean of the ISRFs, the Full Width at Half Maximum (FWHM) was used for the standard deviation $\sigma_{G_0}$ and the amplitude was initialized as $A_{G_0} = (2\pi\sigma_{G_0})^{-1/2}$. For the Super-Gaussian model, the initialization was defined as $\mu_{SG_0} = \mu_{G_0}$, $k_{SG_0} = 2$, $w_{SG_0} = \sqrt{2}\sigma_{G_0}$ and $A_{SG_0} = \frac{k_{SG_0}}{2w_{SG_0}}\Gamma(1/k_{SG_0})$ where $\Gamma$ is the gamma function. The algorithm was stopped after a maximum number of iterations equal to 20000. The dictionary used by SPIRIT was constructed using an SVD of a collection of approximately 10% of the total number of ISRFs within the band of interest, or estimated using the K-SVD algorithm initialized with this collection. In our experiments, we used $N_D = 25$. Two different sparse coding methods based on LASSO (Tibshirani, 1996) and OMP are investigated after dictionary construction. The first method uses a MATLAB implementation of LASSO with a parameter $\mu > 0$ adjusted to obtain a desired number of atoms. The non zero coefficients obtained with LASSO were re-estimated in order to reduce the shrinking bias inherent to this method (Zhang and Huang, 2008). The implementations of the OMP and LASSO algorithms are summarized in Appendices A3 and A4.

## 5.2 ISRF estimation performance

### 5.2.1 ISRF estimation for the MicroCarb mission

An example of ISRF simulated for the MicroCarb mission, and the estimates obtained with the different methods, are displayed in Fig. 4. The results clearly illustrate the advantage of using SPIRIT for ISRF estimation, which leads to normalized estimation errors of less than 1%, significantly below those obtained using the parametric estimation methods. A comparison between the different sparse approximations (OMP, LASSO) and dictionaries (SVD, K-SVD) that can be used by SPIRIT shows that OMP works better than LASSO for this example. Also, using the K-SVD algorithm does not significantly improve the results with respect to SVD, although it has significantly higher computational complexity.

The spectral measurements displayed in Fig. 5 were simulated by the CNES for the B1 wavelength range (758.4-768.9 nm). Results show that, for the MicroCarb spectrometer, the use of the Super-Gauss parameterization reduces the residual error and ISRF approximation errors compared with the Gaussian model. SPIRIT yields significantly better results, with ISRF approximation errors below 1%, and of the order of 0.1% for certain wavelengths. LASSO leads to overall less accurate approximations of the ISRFs than OMP, at significantly higher computational cost, and the use of OMP is overall and consistently beneficial.

**Sum of two generalized Gaussians.** ISRFs can also be modeled using other parametric models, such as the sum of two generalized Gaussians with different shifted center wavelengths, although this has not yet been reported in the literature. As displayed in Fig. 6, this novel parametric approach yields enhanced outcomes as compared to the use of Gaussian and Super-Gaussian models. However, the performance is still not competitive with respect to sparse representation-based methods and

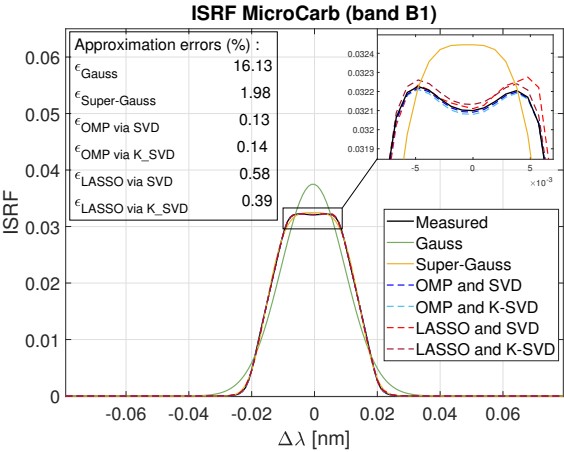

**Figure 4.** Example of a simulated ISRF for the MicroCarb mission and its estimates using parametric methods and SPIRIT.

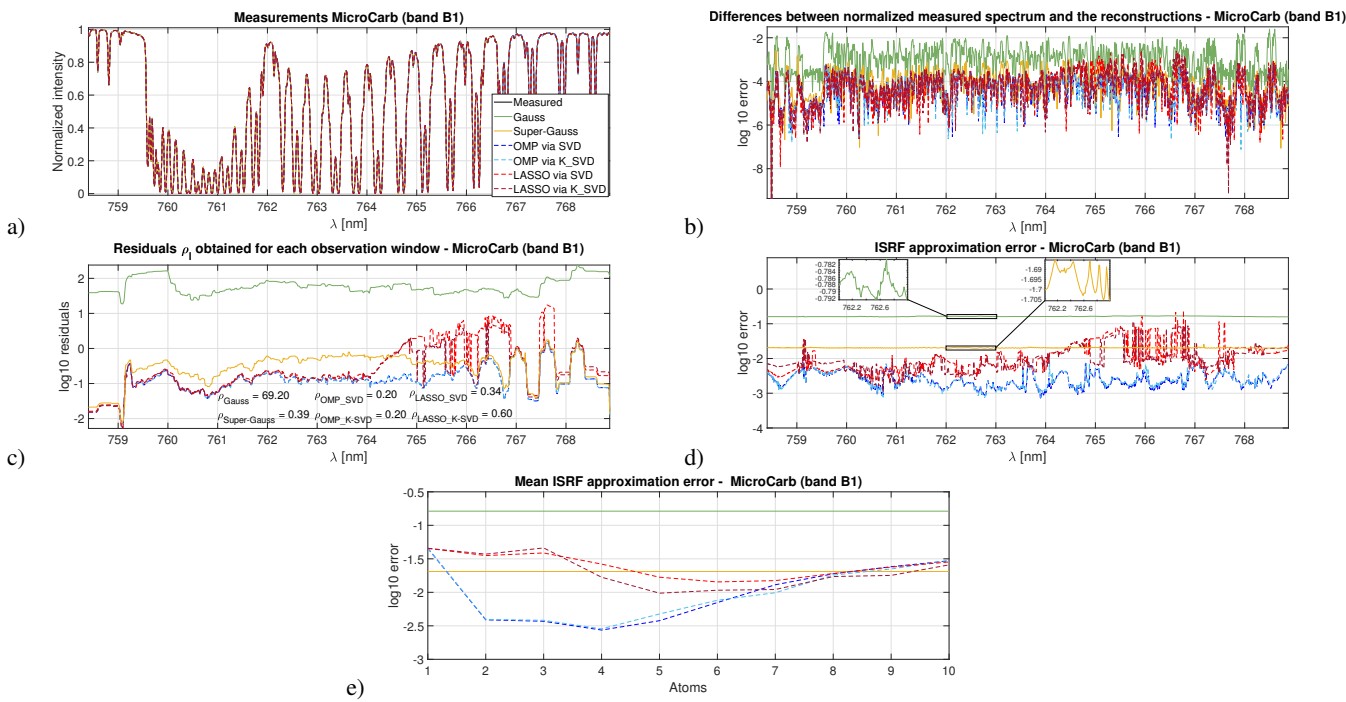

**Figure 5.** Illustrations of a) the measured spectrum reconstruction, b) the difference between the measured spectrum and the reconstructed ones, c) the residuals $\rho_l$ for each wavelengths, d) the ISRF approximation error versus the wavelength and e) the mean ISRF approximation error versus the number of selected atoms for different methods (Gauss, Super-Gauss, OMP, LASSO, SVD and K-SVD) and for the band B1 of the MicroCarb instrument.

necessitates more parameters to estimate. A more detailed study of such more complex parametric models is left for future
work.

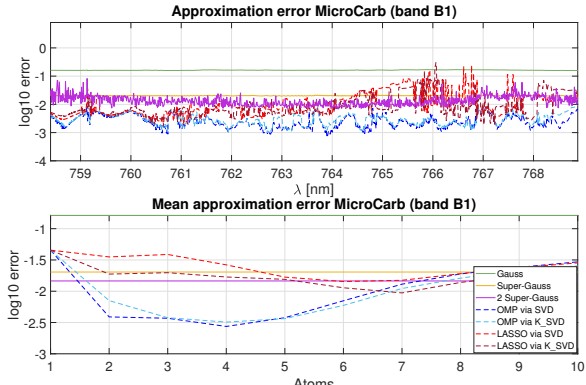

**Figure 6.** Results obtained using the different methods with a dictionary constructed using 103 ISRFs from the band B1 for the sparse
representation-based methods.

### 5.2.2 ISRF estimation for the Orbiting Carbon Observatory 2 (OCO-2) spectrometer

This section studies the applicability of the proposed method to the Orbiting Carbon Observatory 2 (OCO-2) spectrometer.
The OCO-2 spectrometer is used in a NASA Earth observing satellite mission that was launched in July 2014. This mission is
230 dedicated to the study of atmospheric carbon dioxyde and oxygen and aims at characterizing the global $CO_2$ seasonal cycles
and to quantify the sources and sinks of carbon. OCO-2 is composed of three high spectral resolution imaging spectrometers
for narrow spectral ranges. The characterization of ISRFs for this spectrometer is highly challenging and crucial due to this
high spectral resolution. The ISRFs are measured for each pixel using a tunable diode laser during pre-flight calibration (Lee
et al., 2017), and the results are stored in a look-up table. The data used in this article can be downloaded on the NASA data
website EarthDATA (OCO-2 Science Team / Gunson and Eldering, 2019) [6]. The product considered in this study is the OCO-2
Level 1B Version 11r for science acquired in March 2023 and the fourth footprint is used. Specification on the data product can
be found in (Crisp et al., 2021). Some of the ISRFs are declared as unvalid due to radiometric, spatial, spectral or polarization
problems (and are thus not considered for ISRF estimation). The ISRFs associated with bad pixels have not been considered in
our experiments, resulting in a number of ISRFs lower than the number of pixels. To identify the ISRFs at the missing nominal
wavelengths $\lambda_l$, a linear interpolation between two specified nominal wavelengths $\lambda_a$ and $\lambda_b$ with known ISRFs was employed.
The resulting interpolated ISRF is defined by:

$$I_l = \frac{\lambda_l - \lambda_a}{\lambda_b - \lambda_a} I_b + \frac{\lambda_b - \lambda_l}{\lambda_b - \lambda_a} I_a. \tag{9}$$

---

[6]Data available at https://disc.gsfc.nasa.gov/datacollection/OCO2_L1B_Calibration_11r.html.

Note that the number $N_\lambda$ of wavelengths after interpolation may differ from the number of pixels of the instrument, which occurs if the ISRFs associated with the first and/or last pixels are missing. The ISRFs used for the experiments come from the $O_2$A-band of OCO-2 with $N_\lambda = 859$ ISRFs and a sample size $N = 895$. Fig. 7 displays an example of ISRF from the OCO-2 dataset. A visual comparison with Fig. 5 shows that the ISRF shapes can differ significantly depending on the considered wavelength and the instrument. This observation suggests that the dictionary must be adapted to the spectrometer. Another interesting observation is that although the Super-Gaussian distribution should theoretically always provide a better fit than the Gaussian distribution, it is not systematically the case in practice because of convergence issues for the iterative methods used to solve the nonlinear least squares problem for parameter estimation. Specifically, the model parameters are estimated using a simplex-based optimization method (MATLAB function fminsearch) that aims at minimizin the residuals between the measured and estimated spectra, which does not always converge to a better solution for the Super Gaussian model than for the Gaussian model.

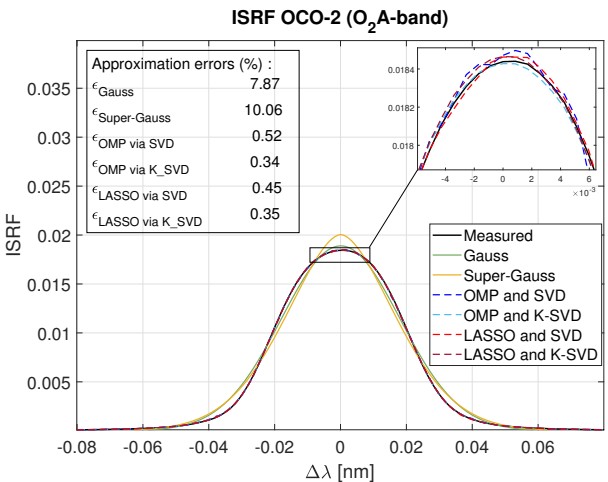

**Figure 7.** Example of an ISRF retrieved for the OCO-2 mission and its estimates using parametric methods and SPIRIT.

Fig. 8 displays performance results for the OCO-2 measurements obtained using the data for the $O_2$ band (757-772 nm). The measured spectrum is reconstructed with the proposed sparse representation methods for $K = 5$ atoms chosen using a dictionary constructed using SVD or K-SVD. The results indicate that the Super-Gaussian model delivers slightly better results than the Gaussian model in terms of residual error and mean ISRF approximation error. However, for the smaller wavelengths of the band, the ISRF approximation errors are slightly larger with the Super-Gaussian model, as already observed in Fig. 7 for a single ISRF. Both parametric models yield close to $10\%$ ISRF approximation errors. The proposed sparse representation approach again yields far better ISRF approximations and measurement fits, with the best results obtained using OMP and SVD.

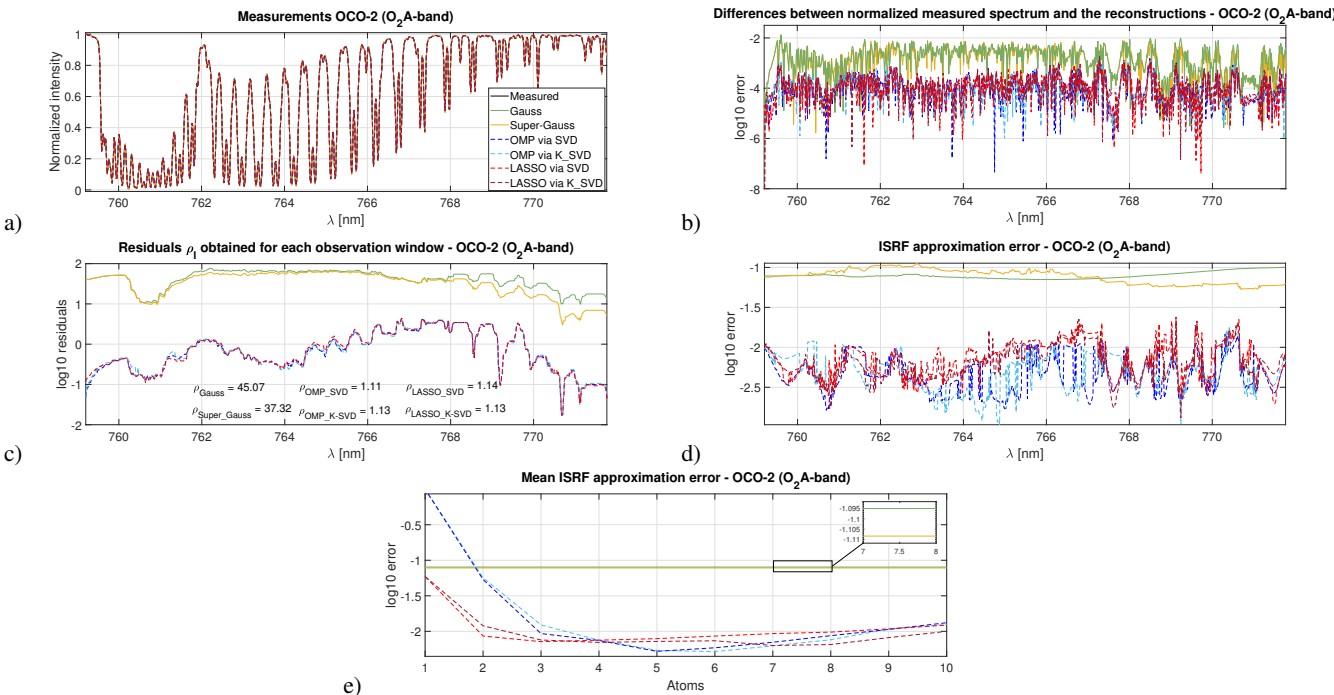

**Figure 8.** Illustrations of a) the measured spectrum reconstruction, b) the difference between the measured spectrum and the reconstructed ones, c) the residuals $\rho_l$ for each wavelength, d) the ISRF approximation error versus the wavelength and e) the mean ISRF approximation error versus the number of selected atoms using different methods (Gauss, Super-Gauss, OMP and LASSO with SVD or K-SVD) for the $O_2$A-band of the OCO-2 instrument.

### 5.2.3 Conclusions

Overall, the conclusions from these experiments are as follows. First, the Super-Gaussian parameterization often yields better performance than the Gaussian one, corroborating the results reported in (Beirle et al., 2017). However, the normalized ISRF approximation errors obtained with these parametric methods are consistently larger than $1\%$, for both instruments and for all wavelengths. In contrast, the proposed SPIRIT approach based on sparse approximations of ISRFs in a suitable dictionary yields significantly better results. This result is due to the fact that the ISRF shapes depend strongly on the spectrometer and can vary across wavelengths, which cannot be accommodated easily with a simple parametric model. On the contrary, decompositions in appropriate dictionaries that depend on the spectrometer and the chosen wavelength offer sufficient flexibility for all use cases considered in this paper. Regarding the estimation algorithms, SVD overall provides an estimation performance close to K-SVD and OMP leads to better estimation than LASSO. There is no theoretical reason for OMP to provide better performance than LASSO. However, it is important to note that the OMP and LASSO algorithms address two distinct problems: the OMP algorithm provides an approximate solution to the problem with an $\ell_0$ penalty and the LASSO algorithm solves the relaxed problem using an $\ell_1$ regularization. Certain limitations of the LASSO algorithm have been highlighted in numerous

publications including (Tibshirani, 1996), and may also be at the origin of our observation. The results overall suggest the use of SVD for building the dictionary and OMP for ISRF estimation.

The proposed methods can also be applied to other instruments, such as Avantes, GOME-2, OMI and TROPOMI used in (Beirle et al., 2017). Results obtained with these instruments are available in the supplementary material (El Haouari et al., 2024) and lead to similar conclusions.

### 5.3 Robustness analysis and ablation study

#### 5.3.1 Robustness to additive noise

Monte Carlo simulations were conducted to study the robustness of the different ISRF estimation methods to the presence of measurement noise. Independent white Gaussian noise was added to the spectral measurements with several signal to noise ratio (SNR) levels to take into account thermal noise and spatial binning: Spatial binning involves the arbitrary division of the
285 imaged area on Earth into distinct field of views (FOVs) (e.g., three FOVs for MicroCarb). The measured spectrum for each FOV is obtained as an average of the measured spectra within that FOV. [7] Table 1 reports the obtained residual approximation errors and the normalized average ISRF approximation errors for the two instruments MicroCarb and OCO-2. Approximation errors less than $< 1\%$ are highlighted in blue. These results show that the proposed sparse representations meet this target for SNRs larger than 20dB. Moreover, OMP is found to be more robust to noise than LASSO and yields overall best results. The
290 parametric models again lead to large errors. It is interesting to note that these errors do not vary significantly with the noise level. This indicates that errors due to model misfit are larger than those induced by the noise degradations. To conclude, OMP combined with SVD provides the overall best results for ISRF estimation, also in the presence of additive noise.

#### 5.3.2 Sensitivity to parameter tuning for SPIRIT

The proposed approach requires the choice of a small number of parameters, namely the size of the sliding window $N_{\text{obs}}$, the
295 size of the dictionary $N_{\text{D}}$ and the number of atoms $K$. The choice of $K$ has been studied above and the best results were obtained for $K \approx 4 - 5$ for both instruments, see Figs. 5 and 8 and the corresponding discussions in Section 5.2.2. Here, we further study the impact of $N_{\text{obs}}$ and $N_{\text{D}}$ on the ISRF approximation errors. To this end, Figs. 9 and 10 show the approximation errors (in $log_{10}$ scale) as a function of $N_{\text{obs}}$ for the Gaussian and Super-Gaussian parameterizations, and as functions of $(N_{\text{obs}}, N_{\text{D}})$ for SPIRIT. Results are reported for the two instruments OCO-2 and MicroCarb and averaged for all ISRFs. The
300 ISRF estimation errors decrease as $N_{\text{obs}}$ increases, as expected. However, this decrease is more important for SPIRIT (e.g., for $N_{\text{obs}} = 80$, the mean ISRF errors for Gauss and Super-Gauss are equal to 16.27%, 2.04%, whereas they are equal to 0.29% for OMP/SVD, 0.33% for OMP/K-SVD, 1.23% for LASSO/SVD and 1.40% for LASSO/K-SVD) showing the interest of exploiting sparsity for ISRF estimation. The results in Figs. 9 and 10 also indicate that it is beneficial to use dictionaries of

---

[7] In the case of the MicroCarb mission, the binning represents a compromise between the objective of achieving a good signal-to-noise ratio (SNR) and maintaining a suitable ground grid, which has a resolution of 13.5 km in ACT and 9 km along the track.

**Table 1.** Mean residual and approximation errors for different SNRs and different methods (Gauss (G), Super-Gauss (SG), OMP and LASSO, SVD and K-SVD).

| Instrument / SNR | Mean ISRF approximation error (%) | | | | | | Residual error | | | | | |
|---|---|---|---|---|---|---|---|---|---|---|---|---|
| | G | SG | OMP SVD | OMP K-SVD | LASSO SVD | LASSO K-SVD | G | SG | OMP SVD | OMP K-SVD | LASSO SVD | LASSO K-SVD |
| 20 dB | 16.28 | **3.39** | 4.58 | 4.37 | 14.38 | 14.23 | 185.5 | 116.4 | **112.3** | 112.4 | 112.7 | 112.9 |
| MicroCarb 40 dB | 16.27 | 2.04 | **0.54** | 0.56 | 2.05 | 2.37 | 70.2 | 1.56 | **1.32** | **1.32** | 1.53 | 1.70 |
| band B1 55 dB | 16.27 | 2.03 | **0.29** | 0.33 | 1.33 | 1.66 | 69.21 | 0.43 | **0.23** | 0.24 | 0.38 | 0.61 |
| 80 dB | 16.27 | 2.03 | **0.28** | 0.32 | 1.27 | 1.68 | 69.2 | 0.39 | **0.20** | **0.20** | 0.34 | 0.60 |
| 20 dB | 8.11 | 8.10 | 4.50 | **3.98** | 6.58 | 5.82 | 174.6 | 165.1 | 121.2 | 122.0 | **120.8** | 121.8 |
| OCO-2 40 dB | 8.04 | 7.80 | 0.79 | **0.74** | 1.12 | 0.96 | 46.35 | 38.60 | **2.33** | 2.35 | 2.34 | 2.35 |
| band 1 55 dB | 8.03 | 7.79 | **0.54** | 0.56 | 0.84 | 0.76 | 45.11 | 37.36 | **1.15** | 1.17 | 1.18 | 1.16 |
| 80 dB | 8.03 | 7.79 | **0.52** | 0.55 | 0.83 | 0.73 | 45.07 | 37.32 | **1.11** | 1.13 | 1.14 | 1.13 |

modest size, since the ISRF estimation errors increase for large dictionaries ($N_D \leq 100$ for OMP and $N_D \leq 25$ for LASSO). Based on this observation, $N_D = 25$ was used in all the experiments.

### 5.3.3 Robustness to ISRF changes

The ISRFs considered in the previous sections were obtained from uniform scenes referred to as "ISRF IN" for the MicroCarb mission. However these ISRFs can change depending on the scene observed by the instrument.

**ISRFs for non-uniform scenes.** The design of the MicroCarb instrument makes the ISRF sensitive to the slit illumination during the integration time. Such dependence on the scene can impact a multitude of instruments. [8] Eight different scenes of the Earth's surface that are directly observed by the spectrometer's slit and subsequently recorded by the instrument's detector during the integration period are considered and are displayed in Fig. 12. These images were obtained in the ACT direction and each image was divided along the ACT direction into three equal parts, resulting in three defined FOVs, labeled as FOV1, FOV2 and FOV3. The spatial pixels in each FOV are averaged to increase the spectral SNR. This binning and averaging step allows three measured spectra per imaged area to be determined, whose ISRFs have to be estimated. Figure 11 shows ISRFs from uniform scenes (left) randomly selected out of the 1024 ISRFs, and ISRFs from non-uniform scenes (right), randomly selected from the total set of eight scenes and three FOVs, highlighting the differences in ISRF shapes depending on the scene: The ISRFs can be more asymmetric for non uniform scenes and are thus harder to estimate. [9] It is interesting to note that the

---

[8]It can be possible to defocus the instrument in order to avoid this dependence on the slit illumination. However, the introduction of a defocus can potentially compromise the precision of the instrument, and thus it was ultimately decided to exclude this option for the MicroCarb instrument.

[9]In practice, there is no information available regarding the non-uniformity of a given scene from the measured spectra. It is only during the inversion process, when estimating the ISRFs, that it becomes apparent (by looking at the measured spectra and the associated residuals) that the ISRFs have been modified. For a given reference spectrum, non-uniform scenes are generated using asymmetric ISRFs, see (Pittet et al., 2019) for more details.

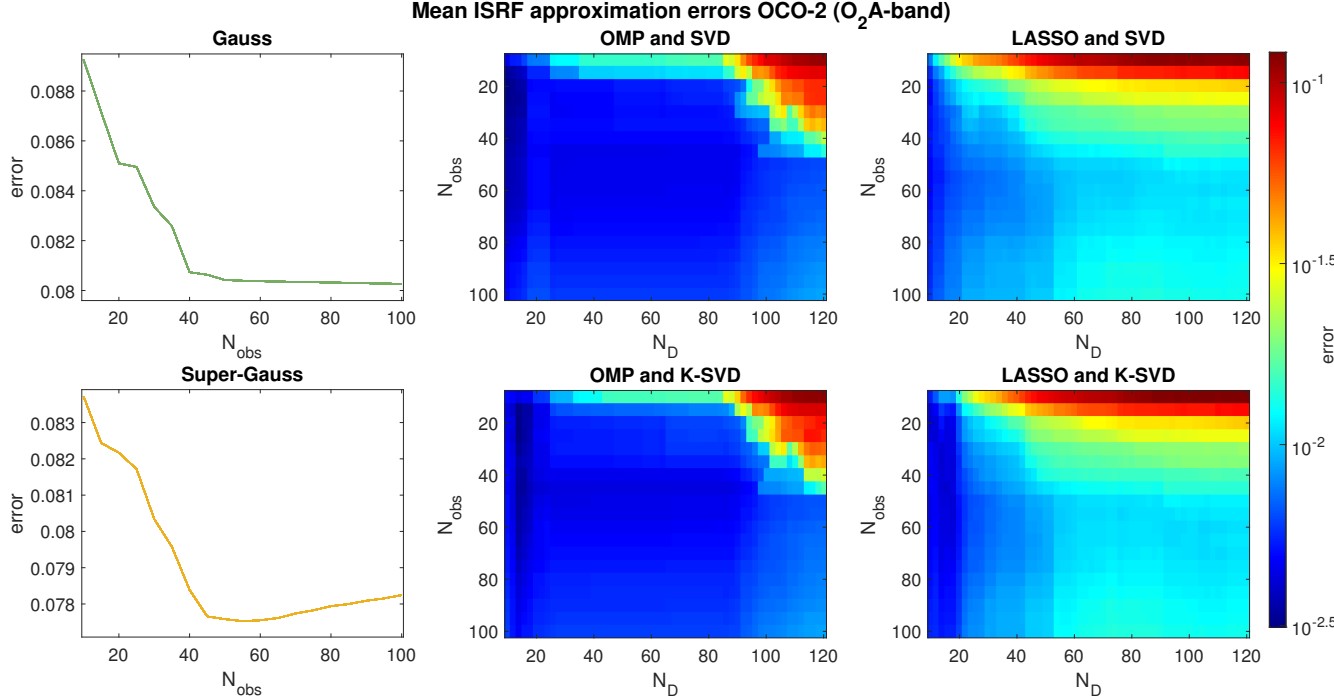

**Figure 9.** Mean approximation errors for OCO-2 and the different estimation methods (Gauss, Super-Gauss, OMP and LASSO with SVD or K-SVD) versus the number of observations $N_{obs}$ and the dictionary size $N_D$ for K = 5.

ISRF of a desert scene is very similar to the ISRF of a uniform scene, contrary to the ISRF of a horizontal coast profil, which
makes the slit blinded during one third of the integration time and leads to an asymmetric left-distorted ISRF, which is harder
to estimate.

**Estimation performance.** This section studies the performance of SPIRIT for estimating non uniform scene ISRFs for the
first band (band B1) of the MicroCarb spectrometer. Two cases are considered: estimation using the original dictionary learned
from examples of uniform ISRFs (ISRF IN), and estimation after modification of this dictionary to account for the diversity
of ISRFs. Specifically, the second dictionary is constructed from from a set of $103$ ISRFs IN (one out of ten) and of $3$ ISRFs
Scene (out of 24). The second dictionary is then composed of $N_D = 25$ new atoms obtained by SVD from this collection of
representative ISRFs. Results obtained using SPIRIT with OMP are displayed in Fig. 13. In the first case (dictionary learnt
from uniform ISRFs, Fig. 13 top row), the resulting normalized ISRF errors exceed 1 % for several scenes and FOVs, pointing
to the fact that the dictionary is not well adapted for representing ISRFs for non-uniform scenes. The results obtained using
the second dictionary are presented in the bottom part of Fig. 13. Using only three additional examples of ISRFs Scene in the
dictionary allows ISRF estimation errors to be again smaller than 1%. Note that the lowest approximation errors are obtained
in most cases using $K = 3$ to $K = 6$ atoms from the dictionary, as before. To conclude, these results show that the proposed

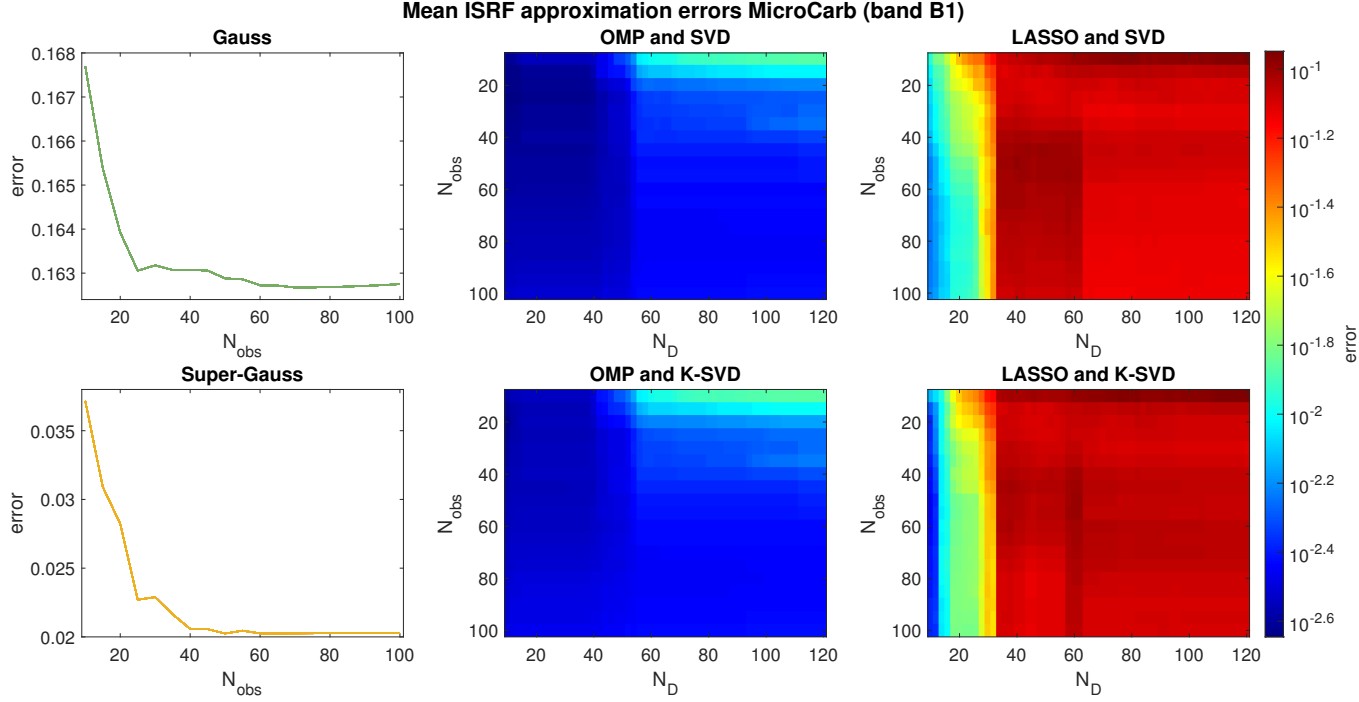

**Figure 10.** Mean approximation errors for MicroCarb and the different estimation methods (Gauss, Super-Gauss, OMP and LASSO with SVD or K-SVD) versus the number of observations $N_{obs}$ and the dictionary size $N_D$ for K = 4.

method can easily adapt to more complex ISRF shapes by considering more diverse ISRF examples in the dictionary estimation step.

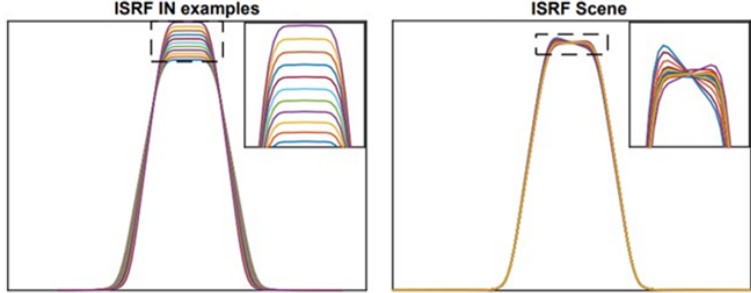

**Figure 11.** Examples of ISRFs from uniform scenes (ISRF IN - left) and from different non-uniform scenes displayed in Fig. 12 and FOVs (ISRF scene - right) (MicroCarb band B1).

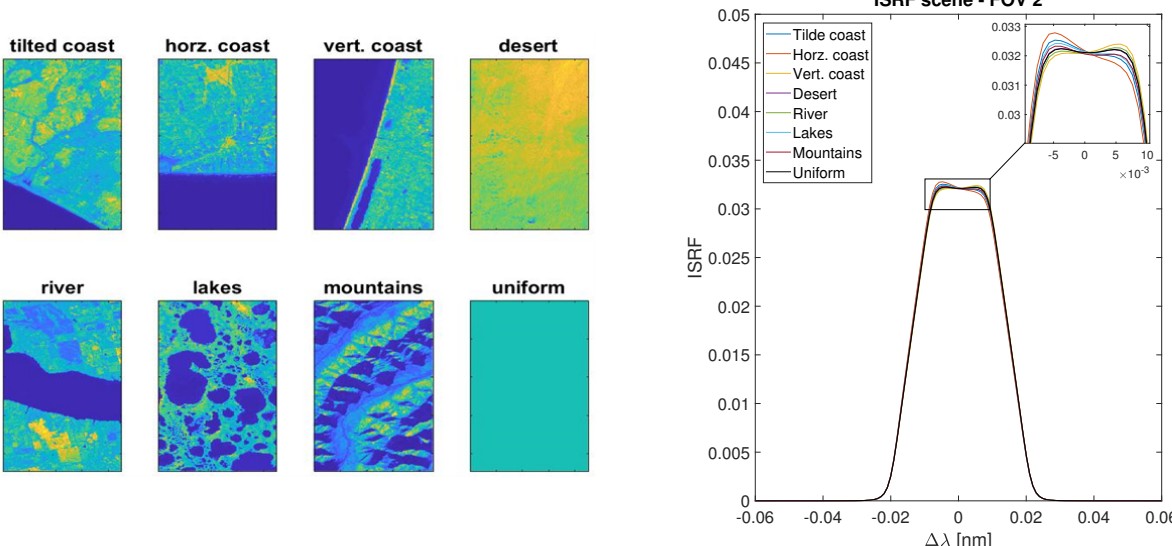

**Figure 12.** Eight types of scenes (left) with the corresponding ISRFs (FOV 2) (right) for the MicroCarb instrument.

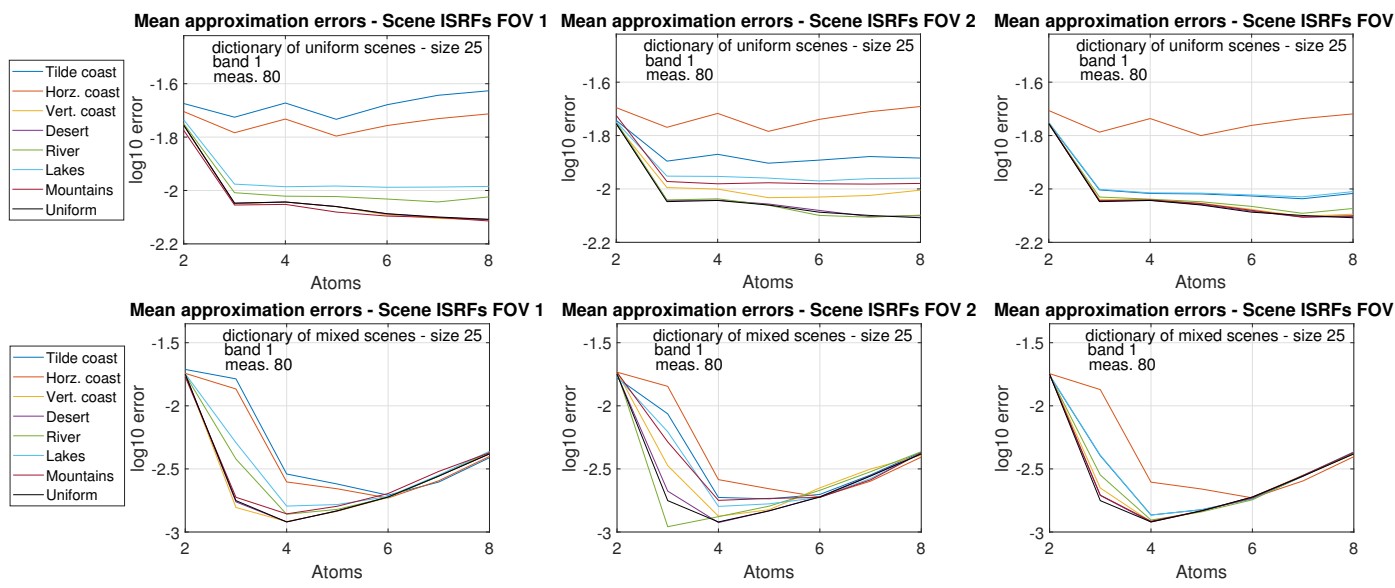

**Figure 13.** ISRF estimation errors for ISRFs Scene obtained using a dictionary of uniform ISRFs (top) and mixed ISRFs (bottom).

### 5.3.4 Robustness to pixel errors

Instrumental errors within a single pixel $l$ can distort the shape of the ISRF of this pixel, leading to the creation of an outlier. This section investigates the impact of such outliers on ISRF estimation. To simulate this scenario, an ISRF from the OCO-2

instrument was inserted in pixel $l = 500$ of the band B1 of MicroCarb data, simulating an outlier in this pixel. The initial
ISRF of the 500th band of Microcarb and its new version are displayed in Fig. 14 (see black and red curves respectively).
The corresponding estimation results, compared to those from the previous study without outliers, are displayed in Fig. 15.
These results demonstrate that the presence of an erroneous ISRF in the sliding window leads to an increase in estimation
errors for the windows containing the outlier since the ISRF estimation becomes more challenging. However, the results also
indicate that the outlier ISRF could be first identified by inspecting the residuals between the measured spectrum and the ISRF
reconstructions and then not considered for ISRF estimation.

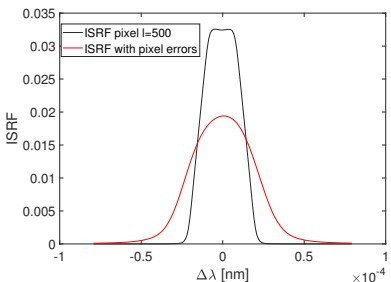

**Figure 14.** Illustration of the generated ISRF (red) at pixel $l = 500$ in presence of pixel errors as compared to the original ISRF (black) for
the band B1 of MicroCarb.

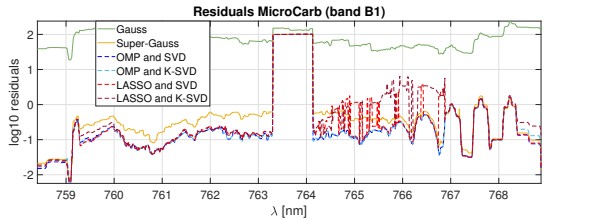 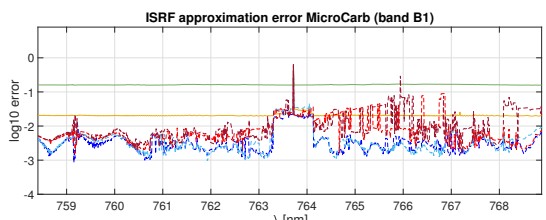

**Figure 15.** Residuals (left) and ISRF estimation errors (right) obtained in presence of pixel errors for the MicroCarb spectrometer using the
different methods (Gauss, Super-Gauss, SVD/KSVD and OMP/LASSO).

**5.3.5 Impact of uncertainties about the reference spectra and reference ISRFs**

This section analyzes the impact of uncertainties about the ISRFs used to build the dictionary or about the reference spectrum
on the ISRF estimation performance.

**Uncertainties about the ISRFs.** To evaluate the impact of uncertainties affecting the ISRFs, Gaussian noise is added to
one-third of the ISRFs used to construct the dictionary, with SNR $= 40$dB and SNR $= 60$dB . The noisy ISRFs are then made
positive by taking their absolute values and normalized to have a unit area. The results, displayed in the left part of Fig. 16
(using $K = 4$ atoms for the plot in the top row), show that as noise increases, better results are achieved with smaller values
of $K$ in the presence of noise with an increase in ISRF estimation errors. However, the estimation is relatively robust to the

presence of noise affecting ISRFs used to build the dictionary since approximation errors remain below 1% on average for both noise levels.

**Uncertainties about the reference spectrum.** In a second experiment, Gaussian noise is added to the reference spectrum, with SNR = 20dB, SNR = 40dB and SNR = 60dB. The results are shown in the right part of Fig. 16 (using $K = 4$ atoms for the plot in the top row). Using a reference spectrum corrupted by additive noise has clearly a smaller impact on estimation performance, when compared to degradations affecting ISRFs used to build the dictionary. Note that high noise levels (SNR = 20 dB) are necessary to significantly increase ISRF estimation errors, probably because of an averaging effect when computing spectral measurement by convolution of the reference spectrum with the ISRF.

Overall, these results indicate that the proposed method is robust to uncertainties in both the ISRFs and the reference spectrum, with ISRF approximation errors remaining below 1% for realistic SNR levels.

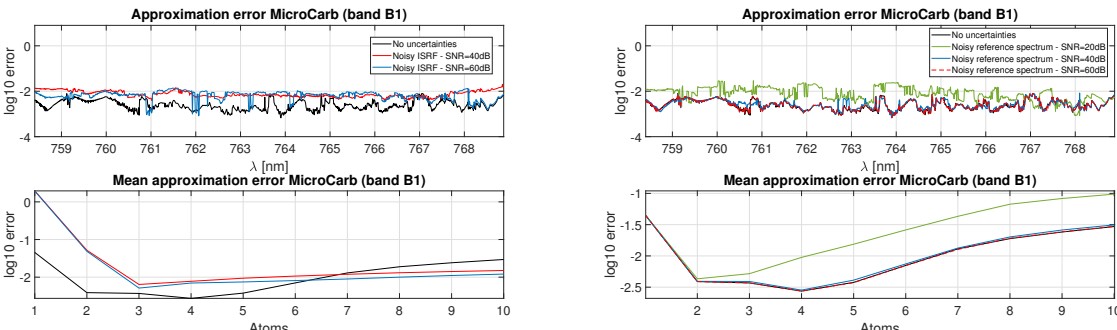

**Figure 16.** Results obtained using SVD and OMP for the different scenarios of noisy ISRFs in the construction of the dictionary (left) and noisy reference spectra (right) for the band B1 of MicroCarb.

## 6 Conclusions

This paper studied a new method for estimating the instrument spectral response functions (ISRFs) of spectrometers. This method is based on a sparse decomposition of the ISRFs into a dictionary of basis functions called atoms. The proposed method can be applied to a large variety of instruments as long as the ISRF estimation problem can be formulated as a linear inverse problem with a sufficient number of measurements (either because the ISRFs do not vary much in a small observation window, in the spectral or spatial domains, or because observations from several reference spectra can be obtained for the same ISRF). The method also requires that a sufficient amount and variety of reference ISRFs have been identified and characterized on the ground to construct the dictionary. We recommend to use the SVD algorithm to build the dictionary using representatives ISRFs and the orthogonal matching pursuit (OMP) algorithm to decompose the ISRFs into this dictionary. The performance of these algorithms is excellent at the price of a very modest computational cost, which suggests its practicality for in-flight scenarios. Another interesting property of the proposed estimation method is that it is not impacted significantly by the shapes of the ISRFs to be estimated, allowing accurate estimations for different types of scenes. Numerical experiments presented in

this paper also showed that the ISRFs of MicroCarb and OCO-2 spectrometers can be estimated with approximation errors smaller than 1%, which is very promising. Other results available in the supplementary material confirm this conclusion for other spectrometers such as Avantes, GOME-2, OMI and TROPOMI.

Future work includes the consideration of radiometric and spectral errors (such as straylight, residual errors of calibration, temporal drifts or spectral shifts) that can degrade the performance of ISRF estimation. These errors are expected to affect more significantly some specific wavelengths, which suggests to investigate specific algorithms jointly correcting the errors and estimating the ISRFs. The resulting problem is more challenging since there are non-linear relationships between the spectrometer measurements and these radiometric and spectral errors. Another interesting prospect is to analyze the potential interest of other methods, e.g., based on Gaussian mixtures or machine learning algorithms, for error correction and ISRF estimation. Finally, it would be interesting to assess more extensively the impact of potential uncertainties about the reference spectra or the ISRFs used to build the dictionary.

*Code and data availability.* The results obtained with the proposed method for the instruments Avantes, GOME-2, OMI and TROPOMI are provided in the Supplement (El Haouari et al., 2024). More details on the data and code used in this study are available upon request from the corresponding author.

**Appendix A: Algorithms**

Appendix A describes the algorithm used to create the matrix of reference spectra, the OMP algorithm and the K-SVD algorithm. The LASSO algorithm was implemented using the Matlab function *lasso.m*. The method used to select the hyperparameter $\mu$ is also presented.

**A1 Reference spectrum matrix**

The algorithm takes as an input the reference spectrum as a vector, the corresponding wavelengths $\boldsymbol{\lambda}_r$, the wavelengths associated with the measured spectrum $\boldsymbol{\lambda}$ and the wavelengths associated with the ISRF $\boldsymbol{\Delta}$ introduced in Section 2.

---
**Algorithm A1** Generation of the reference spectrum matrix.
---

**Input:** Reference spectrum $\boldsymbol{r}$, wavelengths of $\boldsymbol{r}$ denoted as $\boldsymbol{\lambda}_r$, wavelengths of the measured spectrum $\boldsymbol{\lambda}$, ISRF wavelength $\boldsymbol{\Delta}$

**Output:** Reference spectrum matrix for all wavelengths $\boldsymbol{R}$.

1: **for** $l = 1, ..., N_\lambda$ **do**
2:      $\lambda_l = \boldsymbol{\lambda}(l)$
3:      $\boldsymbol{\lambda}_{\text{resp}} = \lambda_l + \boldsymbol{\Delta}$
4:      $\boldsymbol{R}(l,:) = \text{interp}(\boldsymbol{\lambda}_r, \boldsymbol{r}, \boldsymbol{\lambda}_{\text{resp}})$
5: **end for**
6: **return** $\boldsymbol{R}$

---

## A2 Construction of the dictionary

This appendix describes the construction of the dictionary $\boldsymbol{\Phi}$ that will be used in the sparse representation-based algorithms K-SVD, LASSO and OMP.

---

**Algorithm A2** Construction of the dictionary.

**Input:** Matrix of selected ISRFs $\boldsymbol{I}$, size of the dictionary $N_{\text{obs}}$
**Output:** Dictionary of ISRFs $\boldsymbol{\Phi}$.

1: $[\boldsymbol{U}, \boldsymbol{\Gamma}, \boldsymbol{V}^*] = \text{SVD}(\boldsymbol{I})$
2: $\boldsymbol{\Phi} = \boldsymbol{V}(:, 1 : N_{\text{obs}})$
3: **return** $\boldsymbol{\Phi}$

---

## A3 OMP algorithm

Appendix A3 describes the OMP algorithm used to find the sparse representation of the ISRF $\boldsymbol{I}_l$ of interest using $K$ non-zero coefficients in the dictionary $\boldsymbol{\Phi}$ from the measured spectrum $\boldsymbol{s}_l$ and the reference spectrum matrix $\boldsymbol{R}_l$ contained in the sliding window.

---

**Algorithm A3** Orthogonal Matching Pursuit (OMP) algorithm.

**Input:** Measured spectrum $\boldsymbol{s}_l$, reference spectrum matrix $\boldsymbol{R}_l$, dictionary of ISRFs $\boldsymbol{\Phi}$, sparsity parameter $K$
**Output:** Sparse vector $\boldsymbol{\alpha}_l$.

1: $\boldsymbol{\Psi}_l = \boldsymbol{R}_l \boldsymbol{\Phi}$
2: $\boldsymbol{U}_1 = \boldsymbol{s}_l$
3: **for** $k = 1, ..., K$ **do**
4:   Find $\Psi_{\gamma_k} \in \boldsymbol{\Psi}_l$ that maximize the scalar product $|\langle \boldsymbol{U}_k, \Psi_{\gamma_k} / ||\Psi_{\gamma_k}|| \rangle|$
5:   Find $[\alpha_{\gamma_1}, ..., \alpha_{\gamma_k}] \in \boldsymbol{\alpha}_l$ that solves $\arg\min_{\boldsymbol{\alpha}} ||\boldsymbol{U}_k - \sum_{k'=1}^{k} \alpha_{\gamma_{k'}} \Psi_{\gamma_{k'}}||_2^2$
6:   $\boldsymbol{U}_{k+1} = \boldsymbol{s}_l - \sum_{k'=1}^{k} \alpha_{\gamma_{k'}} \Psi_{\gamma_{k'}}$
7: **end for**
8: **return** $\boldsymbol{\alpha}_l$

---

## A4 LASSO algorithm

The MATLAB function *lasso.m* is used to find the sparse representation of the ISRF $\boldsymbol{I}_l$ in the dictionary $\boldsymbol{\Phi}$ using $K$ non-
405 zero coefficients, from the measured spectrum $\boldsymbol{s}_l$ and the reference spectrum matrix $\boldsymbol{R}_l$ associated with the sliding window. A dichotomic search is used to obtain the sparsity parameter $\mu$ that leads to a given number non-zero coefficients $K$. The associated algorithm is described in Algorithm A4.

**Algorithm A4** LASSO algorithm.

**Input:** Measured spectrum $\boldsymbol{s}_l$, reference spectrum matrix $\boldsymbol{R}_l$, dictionary of ISRFs $\boldsymbol{\Phi}$, sparsity parameter $K$, mininimum value of the LASSO sparsity parameter $\mu_{\min}$, maximum value of the LASSO sparsity parameter $\mu_{\max}$

**Output:** Sparse vector $\boldsymbol{\alpha}_l$.

1: $\boldsymbol{\Psi}_l = \boldsymbol{R}_l \boldsymbol{\Phi}$

2: $\boldsymbol{\alpha}_{\mathrm{resp}} = lasso(\boldsymbol{\Psi}_l, \boldsymbol{s}_l, \text{'lambda'}, \mu_{\max}, \text{'Alpha'}, 1)$

3: **while** sparsity( $\boldsymbol{\alpha}_{\mathrm{resp}}$ ) $\neq K$ **do**

4:      $\mu = \frac{\mu_{\min} + \mu_{\max}}{2}$

5:      $\boldsymbol{\alpha}_{\mathrm{resp}} = lasso(\boldsymbol{\Psi}_l, \boldsymbol{s}_l, \text{'lambda'}, \mu, \text{'Alpha'}, 1)$

6:      **if** sparsity($\boldsymbol{\alpha}_{\mathrm{resp}}$) $< K$ **then**

7:          $\mu_{\max} = \mu$

8:      **else**

9:          $\mu_{\min} = \mu$

10:      **end if**

11: **end while**

12: Find the non-zero components in $\boldsymbol{\alpha}_{\mathrm{resp}}$ to form the vector $[\gamma_1, ..., \gamma_K]$

13: Re-estimate the non-zero sparse coefficients: Find $[\alpha_{\gamma_1}, ..., \alpha_{\gamma_k}] \in \boldsymbol{\alpha}_l$ that solves $\arg\min_{\boldsymbol{\alpha}} ||\boldsymbol{s}_l - \sum_{k'=1}^{k} \alpha_{\gamma_{k'}} \Psi_{\gamma_{k'}}||_2^2$

14: **return** $\boldsymbol{\alpha}_l$

## A5    K-SVD algorithm

The K-SVD algorithm of (Aharon et al., 2006) is described in Algorithm A5. At each step, the dictionary is updated by
changing its columns separately and sequentially, and applying $K$ singular value decompositions (SVDs) on the appropriate error matrix $\boldsymbol{E}_j$.

**Algorithm A5** Construction of the dictionary using the K-SVD algorithm.

**Input:** Matrix of selected ISRFs $\boldsymbol{I}$, number of selected ISRFs $L$, size of the dictionary $N_{\mathrm{obs}}$, Dictionary $\boldsymbol{\Phi}$ obtained using SVD in Algorithm (2), sparsity parameter $K$

**Output:** New dictionary of ISRFs $\boldsymbol{\Phi}$.

1: **while** not convergence **do**

2:      Sparse coding step: $\boldsymbol{x}_l = \mathrm{OMP}(\boldsymbol{I}_l, \boldsymbol{\Phi}, K) \, \forall \, l = 1, ..., L$

3:      Dictionary update:

4:      **for** $j = 1, ..., N_{\mathrm{obs}}$ **do**

5:          Define the group of examples that uses the j-th colum of the dictionary j, $w_j = \{l | 1 \leq l \leq N, \boldsymbol{x}_T^j(l) \neq 0\}$

6:          Compute the overall representation error matrix, $\boldsymbol{E}_j = \boldsymbol{I} - \sum_{i \neq j} \boldsymbol{\phi}_i \boldsymbol{x}_T^i$

7:          Build $\boldsymbol{E}_j^R$ from $\boldsymbol{E}_j$ using the columns corresponding to $w_j$

8:          SVD decomposition $[\boldsymbol{U}, \boldsymbol{\Gamma}, \boldsymbol{V}^*] = \mathrm{SVD}(\boldsymbol{E}_j^R)$

9:          Update the dictionary column $\boldsymbol{\phi}_j$ as the first column of $\boldsymbol{U}$ and the vector $\boldsymbol{x}_R^j$ as the first column of $\boldsymbol{V}\boldsymbol{\Gamma}(1,1)$.

10:      **end for**

11: **end while**

12: **return** $\boldsymbol{\Phi}$

*Author contributions.* JE gathered the data for different spectrometers and CP for the MicroCarb spectrometer. JMG and CP contributed to a first formalization of the problem. Mathematical formulation, implementation and formal analysis were conducted by JE, JYT and HW. All authors have contributed to the writing process through discussion and feedback.

*Competing interests.* The authors declare that they have no conflict of interest.

*Acknowledgements.* This study was supported by the French Space Agency (CNES), France and by Thales Alenia Space Cannes, France. We would like to thank Denis Jouglet from the department of Atmospheric Sounding at CNES for providing the reference spectra at the different wavelengths. Moreover, we express our gratitude to Steffen Beirle from the Max Planck Institute for Chemistry(MPI-C) for helpful discussions and for providing some ISRF data.

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
