# Peer review of "In-Flight Estimation of Instrument Spectral Response Functions Using Sparse Representations"

_EGUsphere, 2024_

## Author Comment (AC1)

Response to the Community Comment (CC1)

**In-Flight Estimation of Instrument Spectral Response Functions Using Sparse Representations**

Jihanne El Haouari, Jean-Michel Gaucel, Christelle Pittet, Jean-Yves Tourneret, and Herwig Wendt

October 30, 2024

We would like to thank the Editor and the referees and community members for their careful and thoughtful comments about our paper. We really appreciate the time and effort that has gone into their reviews. All the comments have been considered as detailed below. A detailed answer to the comments can be found in the following pages.

**Response to Community Comment (CC1)**

**Comment 1**

The use of an iterative and dictionary-based based approach for estimating ISRFs is a rather original solution.

**Response:**

We thank the community member for his/her positive appreciation of our work.

**Comment 2**

The fact that the most simple method, SVD + OMP, eventually leads to the best performance is a very good, yet somehow surprising, news, and could be further commented: has this to do with a particular choice of the hyper-parameter in (8) or with a lack of discrimination of the L1 norm constraint?

**Response:**

There is, indeed, no theoretical reason for the method SVD + OMP to lead to superior performance when compared to working with the $\ell_1$ norm penalty. In our study, the hyperparameters for the algorithms have been chosen in order to achieve the best results. However, the OMP and LASSO algorithms address two distinct problems. Indeed, when the OMP algorithm provides an approximate solution to the problem with the $\ell_0$ penalty, the LASSO algorithm solves the problem using the $\ell_1$ penalty, hence gives the solution of an alternative, different problem. Certains limitations of the LASSO algorithm have been highlighted in numerous publications, including [1], and may be at the origin of our observations.

**Comment 3**

  The iterations between dictionary estimates and sparse approximation represent an important aspect of the study, and could be further described.

  **Response:**

  Thank you for pointing this out. The following pseudo-codes have been added as an Appendix A in the revision of the manuscript:

- A1. The generation of the matrix of theoretical spectrum.

- A2. The construction of the dictionary.

- A3. The description of the OMP algorithm.

- A4. The description of the LASSO algorithm (for which the MATLAB *lasso* function is used).

- A5. The description of the K-SVD algorithm, which is based on the algorithm defined in [2].
* * *
**Algorithm 1** Generation of the theoretical spectrum matrix.
* * *
**Input:** Theoretical spectrum $\boldsymbol{r}$, wavelengths of the theoretical spectrum $\boldsymbol{\lambda}_r$, wavelengths associated with the measured spectrum $\boldsymbol{\lambda}$, wavelength associated with the ISRF $\boldsymbol{\Delta}$
**Output:** Theoretical spectrum matrix for all wavelengths $\boldsymbol{R}$.

1: **for** $l = 1, ..., N_\lambda$ **do**
2:   $\lambda_l = \boldsymbol{\lambda}(l)$
3:   $\boldsymbol{\lambda}_{\text{resp}} = \lambda_l + \boldsymbol{\Delta}$
4:   $\boldsymbol{R}(l, :) = \text{interp}(\boldsymbol{\lambda}_r, \boldsymbol{r}, \boldsymbol{\lambda}_{\text{resp}})$
5: **end for**
6: **return** $\boldsymbol{R}$
* * *
**Algorithm 2** Construction of the dictionary.
* * *
**Input:** Matrix of selected ISRFs $\boldsymbol{I}$, size of the dictionary $N_{\text{obs}}$
**Output:** Dictionary of ISRFs $\boldsymbol{\Phi}$.

1: $[\boldsymbol{U}, \boldsymbol{\Gamma}, \boldsymbol{V}^*] = \text{SVD}(\boldsymbol{I})$
2: $\boldsymbol{\Phi} = \boldsymbol{V}(:, 1 : N_{\text{obs}})$
3: **return** $\boldsymbol{\Phi}$
* * *
**Algorithm 3** Orthogonal Matching ¨Pursuit (OMP) algorithm.
* * *
**Input:** Measured spectrum $\boldsymbol{s}_l$, theoretical spectrum matrix $\boldsymbol{R}_l$, dictionary of ISRFs $\boldsymbol{\Phi}$, sparsity parameter $K$
**Output:** Sparse vector $\boldsymbol{\alpha}_l$.

1: $\boldsymbol{\Psi}_l = \boldsymbol{R}_l \boldsymbol{\Phi}$
2: $\boldsymbol{U}_1 = \boldsymbol{s}_l$
3: **for** $k = 1, ..., K$ **do**
4:   Find $\Psi_{\gamma_k} \in \boldsymbol{\Psi}_l$ that maximize the scalar product $|\langle \boldsymbol{U}_k, \Psi_{\gamma_k} / ||\Psi_{\gamma_k}|| \rangle|$
5:   Find $[\alpha_{\gamma_1}, ..., \alpha_{\gamma_k}] \in \boldsymbol{\alpha}_l$ that solves $\arg\min_{\boldsymbol{\alpha}} ||\boldsymbol{U}_k - \sum_{k'=1}^{k} \alpha_{\gamma_{k'}} \Psi_{\gamma_{k'}}||_2^2$
6:   $\boldsymbol{U}_{k+1} = \boldsymbol{s}_l - \sum_{k'=1}^{k} \alpha_{\gamma_{k'}} \Psi_{\gamma_{k'}}$
7: **end for**
8: **return** $\boldsymbol{\alpha}_l$
* * ** * *
**Algorithm 4** LASSO algorithm.
* * *
**Input:** Measured spectrum $\boldsymbol{s}_l$, theoretical spectrum matrix $\boldsymbol{R}_l$, dictionary of ISRFs $\boldsymbol{\Phi}$, sparsity parameter $K$, mininimum value of the LASSO sparsity parameter $\mu_{\min}$, maximum value of the LASSO sparsity parameter $\mu_{\max}$

**Output:** Sparse vector $\boldsymbol{\alpha}_l$.

1: $\boldsymbol{\Psi}_l = \boldsymbol{R}_l \boldsymbol{\Phi}$
2: $\boldsymbol{\alpha}_{\mathrm{resp}} = lasso(\boldsymbol{\Psi}_l, \boldsymbol{s}_l, \text{'lambda'}, \mu_{\max}, \text{'Alpha'}, 1)$
3: **while** sparsity($\boldsymbol{\alpha}_{\mathrm{resp}}$) $\neq K$ **do**
4:     $\mu = \frac{\mu_{\min} + \mu_{\max}}{2}$
5:     $\boldsymbol{\alpha}_{\mathrm{resp}} = lasso(\boldsymbol{\Psi}_l, \boldsymbol{s}_l, \text{'lambda'}, \mu, \text{'Alpha'}, 1)$
6:     **if** sparsity($\boldsymbol{\alpha}_{\mathrm{resp}}$) $< K$ **then**
7:       $\mu_{\max} = \mu$
8:     **else**
9:       $\mu_{\min} = \mu$
10:    **end if**
11: **end while**
12: Find the non-zero components in $\boldsymbol{\alpha}_{\mathrm{resp}}$ to form the vector $[\gamma_1, ..., \gamma_K]$
13: Re-estimate the non-zero sparse coefficients: Find $[\alpha_{\gamma_1}, ..., \alpha_{\gamma_k}] \in \boldsymbol{\alpha}_l$ that solves $\arg\min_{\boldsymbol{\alpha}} ||\boldsymbol{s}_l - \sum_{k'=1}^{k} \alpha_{\gamma_{k'}} \Psi_{\gamma_{k'}}||_2^2$
14: **return** $\boldsymbol{\alpha}_l$
* * *
**Algorithm 5** Construction of the dictionary using the K-SVD algorithm.
* * *
**Input:** Matrix of selected ISRFs $\boldsymbol{I}$, number of selected ISRFs $L$, size of the dictionary $N_{\mathrm{obs}}$, Dictionary $\boldsymbol{\Phi}$ obtained using SVD in Algorithm (2), sparsity parameter $K$

**Output:** New dictionary of ISRFs $\boldsymbol{\Phi}$.

1: **while** not convergence **do**
2:     Sparse coding step: $\boldsymbol{x}_l = \mathrm{OMP}(\boldsymbol{I}_l, \boldsymbol{\Phi}, K) \ \forall \ l = 1, ..., L$
3:     Dictionary update:
4:     **for** $j = 1, ..., N_{\mathrm{obs}}$ **do**
5:       Define the group of examples that uses the j-th colum of the dictionary j, $w_j = \{l | 1 \leq l \leq N, \boldsymbol{x}_T^j(l) \neq 0\}$
6:       Compute the overall representation error matrix, $\boldsymbol{E}_j = \boldsymbol{I} - \sum_{i \neq j} \phi_i \boldsymbol{x}_T^i$
7:       Build $\boldsymbol{E}_j^R$ from $\boldsymbol{E}_j$ using the columns corresponding to $w_j$
8:       SVD decomposition $[\boldsymbol{U}, \boldsymbol{\Gamma}, \boldsymbol{V}^*] = \mathrm{SVD}(\boldsymbol{E}_j^R)$
9:       Update the dictionary column $\phi_j$ as the first column of $\boldsymbol{U}$ and the vector $\boldsymbol{x}_R^j$ as the first column of $\boldsymbol{V}\boldsymbol{\Gamma}(1,1)$.
10:    **end for**
11: **end while**
12: **return** $\boldsymbol{\Phi}$
* * *
> **Comment 4**
>
> The adaptation to calibration errors, and temporal drifts of the feature represent high potential perspectives for this work.
>
> **Response:**
>
> Thank you for this important suggestion. We are indeed planning to study how our method could be adapted to calibrations errors, and temporal drifts, and will elaborate on this in the Conclusions and Perspectives of the article.

**References**

[1] R. Tibshirani, "Regression shrinkage and selection via the lasso," *Journal of the Royal Statistical Society (Series B)*, vol. 58, pp. 267–288, 1996.

[2] M. Aharon, M. Elad, and A. Bruckstein, "K-SVD: An algorithm for designing overcomplete dictionaries for sparse representation," *IEEE Trans. Signal Process.*, vol. 54, no. 11, pp. 4311–4322, 2006.

---

## Author Comment (AC2)

Response to the referee (RC1)

**In-Flight Estimation of Instrument Spectral Response Functions Using Sparse Representations**

Jihanne El Haouari, Jean-Michel Gaucel, Christelle Pittet, Jean-Yves Tourneret, and Herwig Wendt

October 30, 2024

We would like to thank the Editor and the referees and community members for their careful and thoughtful comments about our paper. We really appreciate the time and effort that has gone into their reviews. All the comments have been considered as detailed below. A detailed answer to the comments can be found in the following pages.

**Response to Referee (RC1)**

**Comment 1**

The paper focus on the instrument spectral response functions estimation based on sparse methodologies. This is an interesting paper with a new proposed method that reveals to be competitive and outperforms the state of the art, specifically, the parametric based strategies as the Gaussian and Generalized Gaussian.

**Response:**

We thank the referee for his/her appreciation of our work.

**Comment 2**

How confident are you in the reference spectrum obtain using a radiative transfer model and/or the ground characterization for each instrument and what would be the impact of a possible mismatch on such transfer model/ ground characterization?

**Response:**

In practical applications, the instrument is calibrated using the measured spectra for some specific scenes whose reference spectra are well-characterized (Sun, Moon, uniform scenes such as deserts, etc). Illustrations of the impact of potential mismatches have been included in Section 5.4 of the manuscript, including the figures 1, 2 and 3 below. In a first experiment, a Gaussian noise was introduced to the reference spectrum (with signal-to-noise ratios of 20 dB and 80 dB, cf. Fig. 1). In a second experiment, Gaussian noise is introduced into one-third of the ISRFs used to create the dictionary, with signal-to-noise ratios of 40 dB and 80 dB (cf. Fig. 2). As can be observed in the results presented in Fig. 3, when the noise level increases, the errors in approximating the ISRFs increase, as expected. Nonetheless, the overall error remains lower than those obtained when using parametric models. Furthermore, when the dictionary is constructed from these noisy ISRFs, the noise modifies the form of the atoms, thereby degrading the estimation of the ISRFs. However, note that during ISRF on-ground measurements, the laser is of high precision, and such noisy ISRF measurements with SNR = 20dB are unlikely to occur.

A mismatch may also arise if the instrument undergoes motion during flight, resulting in disparate in-flight ISRFs when compared to those characterized on-ground. This phenomenon was partially examined in our work when when we analyzed the ISRF's dependence on the scene. We showed in Sect. 5.4.3 of the paper that incorporating three additional ISRFs from non-uniform scenes is effective to address the mismatch and sufficient to obtain good ISRF estimation performance. Future work related to these issues will be pointed out in the conclusion of the manuscript.

[Figure]

Figure 1: Representation of possible ISRF mismatches by adding different levels of noise (SNR of 40dB or 80dB) versus true associated ISRF.

[Figure]

Figure 2: Representation of possible mismatches regarding the reference spectrum by adding different levels of noise (SNR of 20dB or 80dB) versus true associated reference spectrum.

[Figure]

Figure 3: Results obtained using SVD and OMP for the different scenarios of ISRF and reference spectrum mismatches.

**Comment 3**

What is the number of representatives ISRF examples used in the simulation part? Can you discuss (at least numerically) the minimum number that leads to an error below of the required 1%?

**Response:**

The number of representative ISRFs used in the simulation part depends on the spectrometer. Approximately 10% of the total number of ISRFs within the band was selected to construct the dictionary. This number is sufficient to yield the performances reported in our work.

We would further argue that the question is less about the minimum number of representative ISRFs used to yield good results, but rather about that there is enough diversity in the ISRFs used to construct the dictionary so the atoms thereof are capable of representing the ISRFs encountered in the estimation problem. Indeed, if all the representative ISRFs are taken from the same part of the spectra (end or beginning), performance will degrade. A similar observation is given in Section 5.4.3, taking into account the ISRF dependency on the scene: if the atoms in the dictionary can not properly represent the asymmetric shapes of some ISRFs, the resulting ISRF estimates will be inaccurate.

**Comment 4**

From fig 1 it is clear that the plotted ISRF cannot be accurately modeled by bell-shaped Gaussian distribution nor the generalized Gaussians one. Nevertheless, it seems that a mixture of Gaussians can be a good fit. Can you elaborate more on this option (the number of mixtures can be estimated using BIC or AIC)?

**Response:**

We thank the reviewer for this interesting observation. Indeed, the ISRF could be modeled using a mixture of Gaussians. This approach was not considered in the present article, and the authors are unaware of any previous work on the use of a mixture of Gaussians in this context[a].

One possible drawback of the use of a mixture of Gaussians could be the potentially larger number of parameters to be estimated, contrary to the proposed sparse representations of ISRFs in a dictionary that use only a small number of atoms (typically around four). Nevertheless, we believe that a study of the use of mixtures of Gaussians in this context could be an interesting future research direction. This perspective will be included in the conclusion of the paper.

[a]If the referee has any reference on this approach applied to the ISRF estimation, we would be grateful for the opportunity to conduct a comparative analysis with our proposed approach.

**Comment 5**

The authors said that the analysis of concentrations from two spectrometers could provide a better understanding of the carbon cycle. Nevertheless, it seems that the analysis in the paper has been done separately (ie., per instrument). Is it possible to use data fusion in this case in order to have a more accurate results?

**Response:**

We thank the reviewer for this remark. The associated sentence was misleading, and we corrected it into: "Note that OCO-2 and MicroCarb aim to provide a better understanding of the carbon cycle". The two spectrometers are used for the same purpose. However their design is different and the associated ISRFs to be estimated are different. Thus, the analyses are conducted separately for these two spectrometers.

**Comment 6**

The authors should add a supplementary material, or annex in order to elaborate more on the algorithm, specifically, in page 6, it is not clear how the matrix S is constructed, how we compute the "appropriate" error matrix ... A pseudo code would be much appreciated from the readers

**Response:**

Following the reviewer comment, we have added a pseudo-code for the K-SVD algorithm in Appendix A. The appendix also describes the computation of the associated error matrix, based on the algorithm defined in [1], and is given below.
* * *
**Algorithm 1** Construction of the dictionary using the K-SVD algorithm.
* * *
**Input:** Matrix of selected ISRFs $\boldsymbol{I}$, number of selected ISRFs $L$, size of the dictionary $N_{\text{obs}}$, Dictionary $\boldsymbol{\Phi}$ obtained using SVD in Algorithm (2), sparsity parameter $K$
**Output:** New dictionary of ISRFs $\boldsymbol{\Phi}$.
* * *
1: **while** not convergence **do**
2:    Sparse coding step: $\boldsymbol{x}_l = \text{OMP}(\boldsymbol{I}_l, \boldsymbol{\Phi}, K) \ \forall \ l = 1, ..., L$
3:    Dictionary update:
4:    **for** $j = 1, ..., N_{\text{obs}}$ **do**
5:       Define the group of examples that uses the j-th colum of the dictionary j, $w_j = \{l | 1 \leq l \leq N, \boldsymbol{x}_T^j(l) \neq 0\}$
6:       Compute the overall representation error matrix, $\boldsymbol{E}_j = \boldsymbol{I} - \sum_{i \neq j} \boldsymbol{\phi}_i \boldsymbol{x}_T^i$
7:       Build $\boldsymbol{E}_j^R$ from $\boldsymbol{E}_j$ using the columns corresponding to $w_j$
8:       SVD decomposition $[\boldsymbol{U}, \boldsymbol{\Gamma}, \boldsymbol{V}^*] = \text{SVD}(\boldsymbol{E}_j^R)$
9:       Update the dictionary column $\boldsymbol{\phi}_j$ as the first column of $\boldsymbol{U}$ and the vector $\boldsymbol{x}_R^j$ as the first column of $\boldsymbol{V}\boldsymbol{\Gamma}(1,1)$.
10:    **end for**
11: **end while**
12: **return** $\boldsymbol{\Phi}$
* * *
> ### Comment 7
>
> In page 7, the authors want to assess the robustness of the proposed ISRF. What do you mean exactly by robustness (is it w.r.t. the noise level, a possible mismatch, a possible presence of outliers ... ), can you be more specific?
>
> > **Response:**
> >
> > The use of the word "robustness" in page 7 was indeed misleading. There, we are not assessing the "robustness" of the proposed method w.r.t. noise levels, mismatches, etc., but we are demonstrating the applicability of the method to different spectrometers. The word "robustness" was thus modified to "applicability".

> ### Comment 8
>
> The caption of some figures is too short and needs more explanation, eg, Fig 1, Fig. 2, Fig 9
> Typo in eq 5 (=)
>
> > **Response:**
> >
> > We thank the referee for this comment. The typo was corrected, and the captions of the figures have been modified in order to include more explanations as:
> >
> > - "Examples of MicroCarb ISRFs." becomes "Illustration of all the ISRFs simulated for the band B1 of MicroCarb instrument using uniform scenes."
> >
> > - "Representation of the four first atoms of the dictionary constructed using one SVD (top) or using the K-SVD algorithm (bottom) for the MicroCarb spectrometer (band B1)." becomes "Representation of the four first atoms of the dictionary of ISRFs $\boldsymbol{\Phi}$ constructed using an SVD on the matrix of representative ISRFs (top) or using the K-SVD algorithm using the same matrix of representative ISRFs (bottom) for the MicroCarb spectrometer (band B1)."
> >
> > - "Examples of ISRFs IN, scene and NU (MicroCarb band 1)." becomes "Examples of ISRFs from uniform scenes (ISRF IN - left) and from different non-uniform scenes displayed in Fig. 10 and FOVs (ISRF scene - right) (MicroCarb band B1)."

**References**

[1] M. Aharon, M. Elad, and A. Bruckstein, "K-SVD: An algorithm for designing overcomplete dictionaries for sparse representation," *IEEE Trans. Signal Process.*, vol. 54, no. 11, pp. 4311–4322, 2006.

---

## Author Comment (AC3)

**In-Flight Estimation of Instrument Spectral Response Functions Using Sparse Representations**

Jihanne El Haouari, Jean-Michel Gaucel, Christelle Pittet, Jean-Yves Tourneret, and Herwig Wendt

October 30, 2024

We would like to thank the Editor, the reviewers and the community members for their careful and thoughtful comments about our paper. We really appreciate the time and effort that has gone into their reviews. The comments of the reviewer (RC2) have been considered as detailed below.

A detailed answer to the comments can be found in the following pages and all the reference.

**Response to reviewer (RC2)**

> **Comment 1**
>
> The authors present a novel method to estimate the Instrument Spectral Response Function (ISRF) of pushbroom-like spectrometers using a data driven approach. They propose to model the ISRF as sparse linear combination of an over-complete set (dictionary) of basis functions (atoms) derived from laboratory characterization measurements. The proposed method is applied to estimate the ISRF of one ground-based (Avantes) and 5 space-borne imaging spectrometers (OMI, TROPOMI, GOME-2, OCO-2 and MicroCarb) designed for remote sensing of the Earth's atmospheric composition. The results obtained for these spectrometers are compared to a state-of-the-art reference ISRF model (fit of a generalized Gaussian). The proposed new algorithm outperforms the reference method in all cases presented by the authors.
>
> Accurate post-launch ISRF estimation is a prerequisite for the delivery of accurate atmospheric remote sensing products and the proposed approach is a novel and creative contribution which should be of interest to the community. The presented investigation seems thorough and the main ideas are presented clearly. Despite some minor issues in some equations, the mathematical basis is outlined sufficiently well. The results shown generally support the authors' claims.
>
> > **Response:**
> >
> > We thank the reviewer for his/her appreciation of our work.

**Comment 2**

There are two major issues, which need to be addressed in my opinion:
Firstly, the authors claim (quite correctly), that a continuous post-launch monitoring of the ISRF is required, because instruments change over time (e.g. due to thermal breathing) and because the ISRF is usually scene dependent, because an inhomogeneous along-track illumination of the spectrometer entrance slit leads to a different effective ISRF than the one typically measured during on-ground pre-flight characterization with a homogeneous along track illumination. However, according to my understanding, the training data set used to compile the dictionary of atoms suggested by the authors consists exclusively of laboratory measurements obtained under homogeneous illumination conditions in the laboratory (except for a very limited study for the MicroCarb instrument in chapter 5.4.3). As such, I would expect these measurements to neither contain effects caused by thermal drift nor those caused by inhomogeneous along track illumination (within one pixel). As the usefulness of the proposed method greatly depends on its behavior under these typical conditions, I would recommend to add results obtained with synthetic data (e.g. by changing the FWHM of the ISRFs of the simulated spectra) to analyze the performance of the proposed algorithm under real-world conditions.

**Response:**

We thank the reviewer for this comment. It is true that thermal variations have not been simulated for the data considered in this work. However, it was observed for the MicroCarb mission that the shape of the ISRFs is not very dependent on thermal variations (test at 160K, 165K). Conversely, ISRF variations due to luminance (spectral axis - along track) or according to the observed scene are much more significant and were considered in our work. The proposed methods could thus be applied in the presence of thermal variations.

The effect of inhomogeneous scenes along the track illumination on the MicroCarb instrument is considered in Sect. 5.4.3 of the manuscript. Note that several ISRFs on the spectral axis should be affected by this error, not just one pixel. If, as suggested, an error is made within a pixel, it is assumed to be an error in the instrument's detector, and the main assumption that the ISRF does not change much in a small observation window no longer holds (the variation of ISRF shapes in the window is too far from 1%). A more detailed study of this observation has been added in Sect. 5.4 of the manuscript, where a different ISRF has been generated for a given pixel. As displayed in figure 1, the variation of ISRF shapes in the sliding window of $N_{\text{obs}} = 80$ observations in two scenarios. The first scenario does not introduce any errors in the ISRF shape leading to small ISRF variations in the sliding window. In the second scenario, a different ISRF has been generated for the pixel $l = 500$. As shown in Fig. 2, the introduction of an erroneous ISRF within the sliding window results in an increase of the estimation error. This observation indicates that it is no longer feasible to estimate the erroneous ISRF with a low error. Note that the estimation error and the associated residuals still provide insight into the underlying issue, which is an inherent inaccuracy in the instrument's detector. This sparse representation-based approach can thus also be used to detect instrument-related errors. All in all, the conclusion of this work is that, as long as the observation window assumption is valid and the dictionary is sufficiently diverse, methods based on sparse representation outperform those based on parametric models.

[Figure]

Figure 1: Results obtained for the mean variations of the ISRF shape within the sliding window of 80 observations in the nominal scenario (i.e. no ISRF error) (top) and in case of a detector pixel error (bottom).

[Figure]

Figure 2: Residuals (top) and ISRF estimation errors (bottom) for the scenario of a dectector pixel error (bottom of Fig. 1) using the different methods (Gauss, Super-Gauss, SVD/KSVD and OMP/LASSO.

**Comment 3**

Secondly, I have difficulties understanding, how exactly the training data set was obtained from the laboratory measurements and how the spectra used for validation were generated. Simply stating that the ISRFs were "obtained by spline interpolation" is not sufficient in my opinion without explaining in which dimension and at which nodes the interpolation was carried out.

**Response:**

We thank the reviewer for this comment and have clarified this point as follows. For each instrument, the associated ISRFs were recovered with the corresponding wavelengths. However, the number of ISRFs retrieved, when the ISRFs associated with bad pixels is removed, is less than the number of pixels. Therefore, to identify the ISRF at the remaining nominal wavelengths, a linear interpolation between two specified nominal wavelengths was employed, as recommended in the OMI slit function product (https://www.knmiprojects.nl/projects/ozone-monitoring-instrument/data-products/omslit). The ISRF value at $\lambda_l$ between two given nominal wavelengths $\lambda_a$ and $\lambda_b$ is defined by

$$I_l = \frac{\lambda_l - \lambda_a}{\lambda_b - \lambda_a} I_b + \frac{\lambda_b - \lambda_l}{\lambda_b - \lambda_a} I_a.$$

The ISRFs associated with a wavelength inferior to the first wavelength of the given ISRF or superior to the last wavelength are not retrieved. This information on the generation of the data have been included in Sect. 4 of the revision, when presenting the instrument.

**Comment 4**

Additionally, I am not sure, whether the ISRFs chosen for the validation were part of the training data (modulo noise) or whether additional changes were introduced (e.g. those mentioned above, leading to shapes and FWHMs typically not encountered during on-ground CAL). As I would expect the behavior of the proposed algorithm to depend strongly on the choice of training data (see chapter 5.4.3), this is a critical issue which has to be addressed before publication in my opinion as one strength of the reference (Gaussian) model is its independence of prior knowledge in this regard. Consequently, a fair comparison should investigate these scenarios, which are of high practical importance.

**Response:**

As pointed out by the reviewer, the behavior of the developed method indeed depends on the choice of the training method. In the case of scenes that exhibit minimal variation in ISRFs (e.g., desert scenes), a dictionary constructed from ISRFs associated with these uniform scenes can be employed to identify them. To construct this dictionary, only 10% of the total number of ISRFs are used. The process becomes more intricate when the scenes are no longer uniform since the ISRF shapes are more diverse, including asymmetric variations, as illustrated in Sect. 5.4.3. However, our results demonstrate that incorporating just three additional ISRFs associated with these non-uniform scenes yields quite accurate estimations.

Finally, the statement that the Gaussian and Super-Gaussian models do not require any assumptions regarding the shape of the ISRFs is somewhat misleading to the authors' opinion. Indeed, when using these parametric models, there is a strong prior assumption on the shape of the ISRF (given by the definition of these parametric models), which is, as demonstrated in the article, not always correct. Furthermore, the parametric models yield inferior results in the uniform scenarios, which suggests that their performance in non-uniform scene scenarios may also be limited. The proposed method makes it possible to remove the assumption of Gaussian shape for the ISRFs and to adapt to the shape of the ISRFs in a non-parametric way.

**Comment 5**

Additionally, it is my impression, that the authors have a very comprehensive knowledge of the Micro-Carb mission, which clearly emerges when discussing results related to this specific instrument. Maybe it is worth considering whether the suggested publication could be dedicated entirely to MicroCarb? Demonstrating that the proposed ISRF retrieval method works under a broad variety of conditions for this instrument alone would convince me of its value. I feel, that the studies presented for the other instruments add little substance to an already convincing demonstration in this regard. This might also help to shorten the manuscript and allow to add currently incomplete or ambiguous information.

**Response:**

Our work proposes a new methodology for estimating ISRFs that can be applied to various instruments. Yet, in order to better balance the length and focus of the manuscript, we mainly concentrate our analysis on the MicroCarb instrument, as suggested by the reviewer, and have included only the additional results obtained for a single additional instrument (OCO-2). Since the potential applicability of the proposed method to other instruments (and other designs) is also a significant contribution and a key aspect of this work, showing that it can be employed and outperform parametric models also in other contexts without additional prior information on the instrument or its design, we have moved the results related to the instruments Avantes, GOME-2, OMI and TROPOMI into a supplementary material.

**Comment 6**

2/4: The optical layout described here is basically a pushbroom spectrometer. In principle, other designs are in use as well (e.g. FTIR), which have to be treated differently. Maybe clarify to which spectrometer types your method applies.

**Response:**

It is true that the spectrometers used in this study are passive pushbroom spectrometers (mainly hyperspectral dispersive spectrometers). Indeed alternative designs, such as FTIR, are also employed in practice and are presented in a footnote of Sect. 4.1 (Instruments, datasets & preprocessing) of the paper. These instruments employ a Michelson interferometer, and the ISRFs can be obtained through the inverse Fourier transform. However, this inversion is applied when the optical path difference (OPD) is assumed to be constant, which is not always the case since the OPD may vary depending on the position (rendering the use of the Fourier transform to find the ISRF no longer applicable). Moreover, in certain applications, undersampling may be necessary. In such cases, applying the Fourier transform becomes more challenging. The issue can then be modeled using a linear inverse model, such as the one considered in this study, and the proposed ISRF estimation process can be used. In summary, the proposed method is not limited to specific types of instruments and does not require many details about the instrument. A contribution of this work is that it can be applied to any instrument as long as the problem can be formulated as an inverse problem and the following hypotheses hold:

1. A sufficient number of measurements associated with the same ISRF are available (either because the ISRFs do not vary much in a small observation window, in the spectral or spatial domain, or because observations from several reference spectra can be obtained for the same ISRF).

2. A sufficient amount and variety of ISRFs have been identified and characterized on the ground to construct the dictionary.

**Comment 7**

2/4: Additionally, the telescope creating the virtual image in the slit does not necessarily image the spectrally dispersed scene on the detector (MicroCarb is rather an exception than the rule in this regard). Maybe add a sketch showing the essential principle design you are investigating?

**Response:**

As suggested by the reviewer, the design of the MicroCarb instrument, obtained from [1], was added to the paper as displayed in Fig. 3.

[Figure]

Figure 3: Principle design of the MicroCarb instrument.

**Comment 8**

4/1-4: What exactly do you consider similar w.r.t. the ISRF and how do you formalize this mathematically? The following unnumbered equation seems to suggest that neighboring ISRFs are assumed to be equal (not just similar) inside a window of Nobs bands. Within an accuracy of 1% I would challenge this assumption for the instruments under consideration.

**Response:**

We thank the reviewer for this comment and have clarified this point in the manuscript. The proposed model is indeed applied to ISRFs, which exhibit slight variations along the spectral axis. It is expected that the mean ISRF variation $\frac{1}{N_{\mathrm{obs}}} \sum_{p=-N_{\mathrm{obs}}/2}^{N_{\mathrm{obs}}/2} \sum_{n=-N/2}^{N/2} |I_l(n\Delta) - I_p(n\Delta)|$ for each sliding window $W_l$ will be below 1 %. It can therefore be stated that, in accordance with this hypothesis, the ISRFs are assumed to be equal. However, the larger this variation, the more important the discrepancies in ISRF shapes. This hypothesis is therefore clearly not valid for the whole set of wavelengths. Therefore, we are proposing to define a sliding window whose size has to be adjusted in order to solve the ISRF estimation problem. Alternatively, observations from $N_{\mathrm{obs}}$ reference spectra could be used when available. This equation has been added to formalize the similarity w.r.t. the ISRF in Sect. 5.4.2..

**Comment 9**

Additionally, could you simply solve this problem (a constant ISRF for multiple bands) using Fourier Transform / Wiener Filter without further assumption on the shape of the ISRF ?

**Response:**

Indeed, for some spectrometers (e.g., FTIR), it is possible to use the Fourier Transform or the Wiener Filter to retrieve the ISRFs. However, in the present case, this is not possible because each observation is associated with a single point from a convolution with different portions of the spectrum. Moreover, the number of observations is smaller than the number of unknown ISRF points ($N > N_{\text{obs}}$). The methodology for addressing this type of problem is to either assume a parametric model (e.g., Gaussian or Super-Gaussian models) or a non-parametric model with a sufficiently small number of unknown parameters to be estimated. Our proposed approach solves this problem by making use of sparse representations in a dictionary, which yield competitive results with respect to Gaussian and super-Gaussian models.

**Comment 10**

Why do you need one equation per band (l) instead of a single equation / matrix including all bands simultaneously ?

**Response:**

The OMP algorithm inherently requires independent ISRF estimations, thus leads to solving $N_\lambda$ independent inverse problems (one problem per band). In principle, the problem could also be formulated using a single equation, for all bands jointly. However, this would lead to a significantly more complicated problem and estimation algorithm, which is left for future work. We will outline possible research directions related to this perspective in the manuscript, stating that a prospect for research could be to estimate all the ISRFs simultaneously by introducing or learning a new regularization term to account for the ISRF variation along the wavelengths.

**Comment 11**

4/24-26: It is not obvious to me, why an ISRF model of two (generalized) Gaussians with slightly shifted center wavelengths would be insufficient to model the displayed ISRF. Please elaborate.

**Response:**

The study of an ISRF model using two generalized Gaussians is an interesting suggestion. Based on the reviewer's comment, we have included new results into Section 5.3 of the manuscript: As displayed in Fig. 4, this novel parametric approach yields enhanced outcomes as compared to the classic Gaussian and Super-Gaussian methods. However, the performance of these models is not competitive with respect to sparse representation-based methods. One potential avenue for further investigation would be the use of a mixture of Gaussians and Super-Gaussians for the construction of the dictionary. However, this represents an alternative formulation of the problem that has not yet been explored, and to the best of our knowledge, has not been previously investigated. This perspective will be included in the conclusion of the paper.

[Figure]

Figure 4: Results obtained using the different methods (Gauss, Super-Gauss, OMP, LASSO and SVD, K-SVD) and a combination of two Super-Gaussian.
* * *
**Comment 12**

4/24-26: Are the shown ISRFs obtained under homogeneous illumination or are they part of the "ISRF Scene" examples shown in fig. 9? I would suggest to specify the band / channel / geom. pixel combination for all shown ISRFs to eliminate ambiguity.

**Response:**

We thank the reviewer and we agree that more details about the ISRF presented in Fig. 9 were required. This figure illustrates how the ISRFs can vary depending on the variety of scene types and illuminations. The left ISRFs are representative of uniform scenes, whereas the right ISRFs correspond to non-uniform scenes (the slit was not uniformly illuminated during the integration time). The caption of the figure has been clarified to "Examples of ISRFs from uniform scenes (ISRF IN - left) and from different non-uniform scenes displayed in Fig. 10 and FOVs (ISRF scene - right) (MicroCarb band B1).".

**Comment 13**

Entire Chapter 4: I think more details regarding the generation of the reference spectra is required here:

Which parameters are chosen for the radiative transfer simulations (trace gas concentration profiles, aerosols, scattering, surface albedo, ...)? Which SNR was assumed to generate the noise? Does the SNR change with sensor and/or wavelength? (ref. section 5.4.1) Except from adding noise, are you only using ISRFs included in the training data set (dictionary) or are you also creating reference spectra with ISRFs slightly narrower or wider than the training data (e.g. to simulate sharpening or blurring caused by thermal breathing of the instrument)? If so, how are those modeled ? Do the simulations include the effect of (along-track) surface albedo inhomogeneity within one pixel ? If so, on which length scale / sampling distance ?

**Response:**

We thank the reviewer for this remark. The description of the methodology employed in the generation of the reference spectra have been detailed and clarified: The profiles originate from the Thermodynamical Initial Guess Retrieval (TIGR) database, which is hosted by Aeris data (https://www.aeris-data.fr/en/projects/thermodynamical-initial-guess-retrieval-tigr/). A sample from the database is selected for the generation of the reference spectra. The measured spectra are obtained by convolving the reference spectra with the ISRFs. In order to assess the impact of different types of noise on the measured spectra, a series of Gaussian noise simulations were conducted, the results of which are presented in Section 5.4.1. It would have been possible to model the noise using Poisson noise associated with the luminance and to add an acquisition noise. However, the objective here was to analyze the sensitivity of the different methods to noise. In practice, the method is applied to measured spectra from which the noise has already been reduced (using spatial binning, preliminary estimation).

**Comment 14**

How exactly (along which dimension and at which sampling points) do you interpolate the ISRFs for each instrument? Why does $N_\lambda$ differ from the number of spectral bands for some instruments?

**Response:**

For each instrument, some pixels are referred to as bad pixels or have high errors. The associated ISRFs are thus discarded. Then, as explained in Comment 4, the ISRFs are interpolated between two given nominal wavelengths $\lambda_a$ and $\lambda_b$ for all wavelengths $\lambda_l$ using the following interpolation formula $I_l = \frac{\lambda_l - \lambda_a}{\lambda_b - \lambda_a} I_b + \frac{\lambda_b - \lambda_l}{\lambda_b - \lambda_a} I_a$. For some instruments, $N_\lambda$ differs from the number of spectral bands because no extrapolation was made to retrieve the ISRFs outside the nominal wavelengths.

**Comment 15**

I think the description of each instrument in a separate sub-section does not add a lot of relevant information beyond what can be found in the cited literature. Have you considered summarizing the relevant information (number of spectral bands, source for the ISRFs, reference citation for the instrument/mission) in a table (and remove sections 4.2 to 4.7)? I think this might enhance clarity and readability of the manuscript.

**Response:**

As suggested, only the OCO-2 instrument will be studied in addition to the MicroCarb instrument in the manuscript (see also our answer to Comment 5). The results obtained for the other instruments have been included in a supplementary material. In the proposed revision and in this supplementary material, a table summarizing the information on the different instruments are provided to enhance clarity and readability.

**Comment 16**

10/second equation: Why are you including the entire sliding window into the error measure? Would it not be sufficient and more meaningful to compute the difference between measured spectrum s and simulated spectrum r $\hat{I}$ at the center wavelengths $\lambda_l$ of each channel l? The Nobs nodes left and right of $\lambda_l$ are only used as computational aid as far as I understand and may e.g. increasingly suffer from boundary effects when approaching the limits of the window. Why do you include these effects into the error measure? In order to support a direct comparison with the assumed SNR, I think a relative measure would be desirable as well (ref table 1).

**Response:**

We thank the reviewer and agree that the information presented in the results section (Sect. 5.3) was not sufficiently explained. The second figure simultaneously shows the absolute differences between the normalized spectral measurements and their normalized approximations (in logarithmic form, second rows) and the corresponding mean residual, represented by the symbol $\rho$. To enhance clarity, for each spectrometer, the proposed figures now show the measured spectrum reconstruction, the difference between the measured spectrum and the estimated one, the residuals for each wavelength, which allows the reader to visualize the function which is minimized, the ISRF approximation error and the mean ISRF approximation error versus the number of selected atoms as displayed in Fig. 5. In this study, the ISRF estimation process is conducted through the minimization of an objective function which is defined using the $N_{\mathrm{obs}}$ observations. Displaying the results of the minimization process is important to our opinion for comparison with the outcomes of the alternative methods.

[Figure]

Figure 5: Illustrations of the measured spectrum reconstruction (fig. 1), the difference between the measured spectrum and the reconstructed ones (fig. 2), the residuals for each wavelength (fig. 3), the ISRF approximation error versus the wavelength (fig. 4) and the mean ISRF approximation error versus the number of selected atoms (fig. 5) for the band B1 of the MicroCarb instrument.

**Comment 17**

12/fig(3e): How is it possible, that the Gaussian (as special case of the super-Gaussian) fits the data that much better than the super-Gaussian here? Are you sure the fit converged properly?

**Response:**

It is indeed unexpected that the Gaussian distribution outperforms the Super-Gaussian. The initial insight for answering this question is that, with regard to this particular spectrometer, the ISRFs (illustrated in Fig. 3c) appear to be more Gaussian than the ISRFs of the other spectrometers, allowing a better convergence for the Gaussian model. The second insight is that the problem that is solved, using either the Gaussian or the Super-Gaussian parameterisation, can be described as a nonlinear least-square problem. To address this issue, we employ a simplex-based optimization method (MATLAB function *fminsearch*) to minimize the residuals between the measured and estimated spectra using the parametric models. The super-Gaussian method is initialized using the results obtained with the Gaussian model. This method usually provides better performance than the Gaussian model. However, the simplex-based optimization method does not always converge to the optimal solution, which is the case in this example.

**Comment 18**

How are the ISRFs in this figure normalized (not unit area) ?

**Response:**

In all results, the ISRFs have been normalized to unit area.

**Comment 19**

Chapter 5.2 and 5.3.x: Lacking the information listed above, I feel I cannot comment on the authors' claims here in a meaningful way, as it is of fundamental importance how the reference spectra were chosen and whether they are included in the training data or not.

**Response:**

The authors hope that the information provided on the reference spectra in previous comments (Comment 13, which profiles are obtained using the TIGR database) will help the reader to better understand and appreciate our claims. However, one significant contribution of this work is that as long as the reference spectrum is known, it is possible to estimate the ISRFs with high accuracy. The reference spectrum is supposed well-known (obtained from well-known scenes such as desert, moon, solar scenes) and this spectrum is not included in the training (only ISRFs characterized on the ground are). It is in principle possible to use any known reference spectra with sufficient local variations in its spectral content.

**Comment 20**

Chapter 5.4.1:

Does SNR=50 dB imply SNR=100 000? (which seems quite high to me). In this case I would expect a log10 residual for a "perfect fit" around -5 if sufficiently many atoms are chosen, but the values in fig. 6 (b) are significantly higher. Could you elaborate on this? Looking at table 1, it seems to me that the SPIRIT approach works significantly less effective for SNR = 100 (20 dB), which is not uncommon for many earth observing instruments. Could you comment on the usability of SPIRIT in these scenarios? Chapter 5.4.3: Why do you choose across-track binning into these exact pixel groups? Are the ISRFs for the bands in these geometric regions similar?

**Response:**

The noise considered in our experiments has been generated for different SNR levels. We agree that there is no discernible difference in the results obtained with SNRs of 80dB and 120 dB. Consequently, all results obtained with SNR=120dB have been removed in the revision. In practice, a spatial binning is typically employed to enhance the signal-to-noise ratio (SNR). This binning involves the arbitrary division of the imaged area on Earth into distinct field of views (FOVs) (3 for instance for MicroCarb). The measured spectrum for each FOV is obtained as an average of the measured spectra within that FOV. This is achieved through the application of the following formula: $s(\lambda_l) = \frac{1}{C} \sum_{c=1}^{C} s_c(\lambda_l)$, where $s_c(\lambda_l)$ is the value of the measured spectrum associated with the wavelength $\lambda_l$ at the spatial pixel $c$ in the FOV. It is indeed assumed in this study that the ISRFs do not change much in across-track (spatial axis) as compared to along track (spectral axis). In the case of the MicroCarb mission, the binning represents a compromise between the objective of achieving a satisfactory signal-to-noise ratio (SNR) and that of maintaining a suitable ground grid, which has a resolution of 13.5 km in ACT and 9 km along the track. These comments about the SNR and the binning have been included in Sect. 5.4.1 of the revision, where the robustness to noise is evaluated.

**Comment 21**

How exactly are the ISRFs for inhomogeneous scenes obtained? Any simulation would require knowledge of the ISRFs for an inhomogeneously illuminated pixel/slit I presume. Is this knowledge inferred from measurements or simulations based on an optical instrument model?

**Response:**

For a given reference spectrum, non-uniform scenes are generated using asymmetric ISRFs. In practice, there is no information available regarding the non-uniformity of a given scene from the measured spectra. It is only during the inversion process, when estimating the ISRFs, that it becomes apparent that these ISRFs have been modified. More details can be found in [2].

**Comment 22**

1/2-3: optical elements "induce errors in the measurement"? Then why not leave them out ;-)? Maybe rephrase?

**Response:**

We thank the reviewer for pointing us to this ambiguous phrase. The sentence "Spectrometers are composed of different optical elements that can induce errors in the measurements and therefore need to be modeled as accurately as possible" was modified by "Spectrometers are composed of different optical elements that must be modeled as accurately as possible. In the absence of such precision, the retrieval of trace gas concentrations can be significantly compromised".

**Comment 23**

2/9: Is an ISRF not rather associated with a channel than a wavelength? The (center) wavelength of said channel can then be chosen based on (mean, max or median of) the ISRF?

**Response:**

Indeed, the ISRF is associated with a given pixel, designated by the index "$l$". The wavelength associated with the pixel is then obtained as the center (maximum, median, or barycenter) of the ISRF at the given pixel. However, there are some effects, such as the smile (in ACT) or some gaps in our knowledge about the wavelengths (in along track), that can result in spectral shifts, which can degrade the estimation of ISRFs. The reason behind stating that each pixel is associated with a single ISRF and wavelength is that the influence of these various factors is still under investigation and not accounted for in this work. Consequently, we assume that the wavelength is known and address the ISRF estimation problem by solving an inverse problem. This observation was incorporated in Sect. 2.1 during the ISRF estimation problem as a footnote and the potential impact of spectral shifts was outlined as a future objective of the research.

**Comment 24**

2/14: ... "ISRF wavelength variations exceed this threshold" ? Which wavelength varies here? How does this affect the ISRF error budget ?

**Response:**

This sentence was in fact misleading. The sentence "For some missions, ISRFs are expected to be known with a normalized error less than 1%, which is a challenge since the ISRF wavelength variations exceed this threshold" was modified to "For some missions, ISRFs are expected to be known with a normalized error less than 1%, which represents a significant challenge given that the variations in ISRF shape across the entire band frequently exceed this threshold."

**Comment 25**

2/17: I would argue, that all pushbroom instruments are susceptible to effective ISRF changes if the illumination varies within a pixel (along-track), as partial illumination of the spectrometer entrance slit is equivalent to a narrower slit and thus (usually) a smaller FWHM. Unless optically mitigated (e.g. by means of a slit homogenizer or optical fibers), this effect has to be taken into account in the error budget. Does the 1%-requirement include the associated uncertainties?

**Response:**

It is evident that the ISRF's dependence on the scene can impact a multitude of instruments. This represents a design choice for the instruments. It might have been feasible, for example, for the MicroCarb mission to defocus the instrument in order to avoid this dependence on the slit illumination. However, the introduction of a defocus can potentially compromise the precision of the instrument, and thus it was ultimately decided to exclude this option. The performance of 1% on the ISRF knowledge is a consequence of the MicroCarb mission's necessity for an accurate determination of $CO_2$ concentrations. This performance is an objective of the mission, but it can indeed be challenging to achieve in practice using real data. The 1% requirement accounts for uncertainty, acquisition noise of ISRFs and interpolation. In this study, however, the 1% requirement is a defined threshold for the maximum of the ISRF approximation error.

**Comment 26**

3/9: Technically each channel / geometric pixel combination has an individual ISRF. This ISRF can then be used to define a center wavelength for each channel of each geometric pixel. Have you considered associating each ISRF with the number (l) of a channel instead of the center wavelength? This might simplify the notation in many equations in my opinion.

**Response:**

As discussed in Comment 24, we could define the ISRF using the number ($l$) of the channel instead of the associated wavelength. However, we wanted to model the problem at first sight by being as close as possible to the physical phenomenon behind (in Eq. (1) : $s(\lambda_l) = (r * I_l)(\lambda_l)$), which implies the use of a wavelength. Afterward, the ISRF is defined as a vector using the number $l$ only.

**Comment 27**

3/eq(1): The convolution is usually defined over the entire space of real numbers. Also, the ISRF I is mirrored along the wavelength axes in this notation, as I(u) is the sensitivity to the wavelength $\lambda_l$ - u, which might be unexpected for many readers. It has no practical effect on your results of course.

**Response:**

Based on the reviewer comment, we have changed the notation accordingly.

**Comment 28**

3/eq(2): I think a $\Delta$ is missing in front of the sum. Also: Does $\lambda_l - n\Delta$ equal $\lambda_{l-n}$ ? If not: How do you choose $\Delta$?

**Response:**

It is possible that a misunderstanding has occurred. It is assumed that the ISRFs are defined on a regular grid of wavelengths centered at 0. The associated wavelength gap between two ISRF measurements is designated as $\Delta$. The expression $\lambda_l - n\Delta$ is thus not equal to $\lambda_{l-n}$ given that the space between two points in the ISRF is smaller than that between two points in the measured spectrum. The sentence following Eq. (2) "where $\Delta$ is the wavelength sampling interval for the ISRFs, which is assumed to be regularly sampled." has been modified to: "where $\Delta$ represents the sampling period between two consecutive points of the ISRF, which is assumed to be regularly sampled." and a footnote has been added: "The wavelength grid $\Delta$ represents the points at which the ISRFs are defined".

**Comment 29**

3/21: "A major difficulty with the inverse problem ...": Maybe also mention another very fundamental problem: Eq. 1 is a Fredholm equation, the solution of which is the classical example of an ill-posed problem. Additionally the constraint of a single measurement could be removed experimentally using e.g. the sun and a spectrally tunable on-board calibration source, albeit at extra cost.

**Response:**

This information has been included in the revision next to the denoted sentence.

**Comment 30**

4/eqs(3 & 4): Why $x \in \Delta$ (i.e. the sampling interval) ? Could you not use any real number for x? (For most atmospheric spectrometers the sampling ratio is greater 2, so the FWHM exceeds one sampling interval in most cases.)

**Response:**

It is true that $\Delta$ is the sampling interval. However, $\Delta$ also represents the wavelength grid, i.e., the points where the ISRFs are defined. The equation: "$\Delta = [-\frac{N}{2}\Delta, ..., \frac{N}{2}\Delta]$" was not correct and has been replaced by: "$\Delta = \{-\frac{N}{2}\Delta, ..., \frac{N}{2}\Delta\}$"

**Comment 31**

4/eq(5): I think there is one superfluous equation symbol following the summation symbol and the summation index n should probably occur somewhere in the equation (unless the summation is part of the higher dimensional 2-norm)?

**Response:**

Corrected as suggested.

**Comment 32**

10/5: "carbon" without trailing "e"

**Response:**

Corrected as suggested.

**Comment 33**

10/first equation: If the ISRFs are normalized to unit area, can the denominator have values different from one?

**Response:**

We thank the reviewer for the remark, this equation is the usual expression used to assess the performance of the ISRF estimation method. However, the ISRFs are indeed normalized to unit area. The denominator has been erased and the following has been added next to the equation: "For the selected instrument, the ISRFs are assumed to be normalized to unit area."

**Comment 34**

12/fig(3): The font is barely legible and my aging eyes can hardly discriminate the two LASSO variants. Please increase the size of these plots. The Avantes SVD/K-SVD fits seem to have linear segments, which I would not expect in a "real" ISRF. Are these linear interpolation artifacts? Maybe add markers at the computational nodes for clarity?

**Response:**

The size of the plot has been increased and some markers have been added for SVD/K-SVD fits.

**Comment 35**

13/fig(4): Even at a zoom level of 190 % I can barely read this figure! Please increase the size of this figure and the font size.

**Response:**

We apologize for the lack of readability. The size of the figure has been increased accordingly.

**Comment 36**

20/fig(9): Please indicate for which band / channel / pixel combination these ISRFs are valid.

**Response:**

Figure 9 is composed of two distinct sets of ISRFs. The first set, displayed in the left figure, comprises ISRFs from uniform scenes, randomly selected from the total set of 1024 ISRFs. The second set displayed in the right figure comprises ISRFs from non-uniform scenes, randomly selected from the total set of eight scenes and three FOVs. The figure caption has been modified from "Examples of ISRFs IN, scene and NU (MicroCarb band 1)" to "Examples of ISRFs from uniform scenes (ISRF IN - left) and from different non-uniform scenes displayed in Fig. 10 and FOVs (ISRF scene - right) (MicroCarb band B1)."

**Comment 37**

20/fig(10): Do the figures on the left-hand side show a single pixel IFOV in along-track / across-track direction? Otherwise please indicate the pixel limits. How do these images enter into the ISRFs shown on the right? Are these ISRFs simulated or measured ?

**Response:**

The image displayed in Fig. 10 shows a scene that is directly observed by the spectrometer's slit and subsequently recorded by the instrument's detector during the integration period. This image is then obtained in the ACT direction. From this image, a cut is made along the ACT direction into three equal parts, resulting in three FOVs. The resulting ISRFs have been simulated.

**Comment 38**

17/table 1: I think it would be helpful to indicate the expected minimum error for a given SNR. Considering an ideal Gaussian with synthetic noise at an assumed SNR of 1000 I would e.g. expect a mean residual error of approx. 0.1 %, but I do not observe a relation of this kind in the data. Could you elaborate on the relationship between SNR and normalized approximation error a little bit more in the text?

**Response:**

As suggested by the reviewer, the anticipated minimum error for a specified SNR has been identified in the revision. However, regardless of the method employed, the ISRFs are estimated by using a model. Thus, it is inevitable that their estimation will not be perfect, even in the absence of noise. In the table, the term "normalized approximation error" refers to the discrepancy between the approximated and the actual ISRFs. In order to avoid any confusion, "Normalized approximation error," has been changed to "Mean ISRF approximation error."

**Comment 39**

18/fig(7) & 19/fig(8): Considering the limited range of the error values, I would recommend a linear scale for the y-axis.

**Response:**

Modified as suggested.

**Comment 40**

21/fig(11): This figure is also quite small. Maybe also consider a linear scale. Is it necessary to resolve differences smaller than the noise level?

**Response:**

The size of the figure has been increased. However, it was decided to keep the scale in log10 as for the other results in order to see the differences in the ISRF approximation errors.

**References**

[1] M. Castelnau, E. Cansot, C. Buil, V. Pascal, V. Crombez, S. Lopez, L. Georges, and M. Dubreuil, "Modelization and validation of the diffraction effects in the microcarb instrument for accurately computing the instrumental spectral response function," in *International Conference on Space Optics—ICSO 2018*. SPIE, 2019, vol. 11180, pp. 1054–1068.

[2] C. Pittet, V. Crombez, D. Jouglet, L. Georges, E. Cansot, and A. Albert-Aguilar, "In-flight estimation of the microcarb instrument spectral response functions.," in *Geophysical Research Abstracts*, 2019, vol. 21.

---

## Author Response (AR1)

Response to the Editor, Referees and community members

**In-Flight Estimation of Instrument Spectral Response Functions Using Sparse Representations**

Jihanne El Haouari, Jean-Michel Gaucel, Christelle Pittet, Jean-Yves Tourneret, and Herwig Wendt

December 12, 2024

We would like to thank the Editor and the referees and community members for their careful and thoughtful comments about our paper. We really appreciate the time and effort that has gone into their reviews. All the comments have been considered as detailed below, and the resulting changes appear in red and blue in the revised manuscript.

A detailed answer to the two referee's comments and to the community member comments can be found in the following pages.

**1 Response to the referee RC1 ♯1**

**Comment 1.1**

The paper focus on the instrument spectral response functions estimation based on sparse methodologies. This is an interesting paper with a new proposed method that reveals to be competitive and outperforms the state of the art, specifically, the parametric based strategies as the Gaussian and Generalized Gaussian.

**Response:**

We thank the referee for his/her appreciation of our work.

**Comment 1.2**

How confident are you in the reference spectrum obtain using a radiative transfer model and/or the ground characterization for each instrument and what would be the impact of a possible mismatch on such transfer model/ ground characterization?

**Response:**

In practical applications, the instrument is calibrated using the measured spectra for some specific scenes whose reference spectra are well-characterized (Sun, Moon, uniform scenes such as deserts, etc). Illustrations of the impact of potential mismatches have been included in Section 5.3.5 of the manuscript. In a first experiment, Gaussian noise is added to one-third of the ISRFs used to construct the dictionary, with SNR = 40dB and SNR = 60dB. In a second experiment, Gaussian noise is added to the reference spectrum, with SNR = 20dB, SNR = 40dB and SNR = 60dB. The results displayed in Fig. 16 (page 21) globally show that show that as noise increases, best results are achieved with smaller values of $K$ in the presence of noise, and the error in ISRF estimation increases. Nevertheless, when the noise level increases, the proposed method is robust to uncertainties in both the ISRFs and the reference spectrum, with ISRF approximation errors remaining below 1% for realistic SNR levels.

Note that mismatch may also arise if the instrument undergoes motion during flight, resulting in disparate in-flight ISRFs when compared to those characterized on-ground. This phenomenon was partially examined in our work when we analyzed the ISRF's dependence on the scene. We showed in Section 5.3.3 of the paper that incorporating three additional ISRFs from non-uniform scenes is effective to address the mismatch and sufficient to obtain good ISRF estimation performance. Future work related to these issues will be pointed out in the conclusion of the manuscript.

**Comment 1.3**

What is the number of representatives ISRF examples used in the simulation part? Can you discuss (at least numerically) the minimum number that leads to an error below of the required 1%?

**Response:**

The number of representative ISRFs used in the simulation part depends on the spectrometer. Approximately 10% of the total number of ISRFs within a band was selected to construct the dictionary. This number has been specified in Section 5.1, page 10 and line 203 and is sufficient to yield the performances reported in our work.

We would further argue that the question is less about the minimum number of representative ISRFs used to yield good results, but rather about having enough diversity in the ISRFs used to construct the dictionary so the atoms thereof are capable of representing the ISRFs encountered in the estimation problem. Indeed, if all the representative ISRFs are taken from the same part of the spectra (end or beginning), performance will degrade. A similar observation is given in Section 5.3.3, taking into account the ISRF dependency on the scene: if the atoms of the dictionary cannot properly represent the asymmetric shapes of some ISRFs, the resulting ISRF estimates will be inaccurate.

**Comment 1.4**

From fig 1 it is clear that the plotted ISRF cannot be accurately modeled by bell-shaped Gaussian distribution nor the generalized Gaussians one. Nevertheless, it seems that a mixture of Gaussians can be a good fit. Can you elaborate more on this option (the number of mixtures can be estimated using BIC or AIC)?

**Response:**

We thank the reviewer for this interesting observation. Indeed, it seems that it should be possible to model the ISRF using a mixture of Gaussians or generalized Gaussians. This approach was not considered in the present article, and the authors are unaware of any previous work on the use of a mixture of Gaussians in this context[a]. Based on the reviewer comments, we have included new simulation results in Section 5.2.1 (page 10) showing the interest of using mixtures of generalized Gaussians (sum of two generalized Gaussians), even if the performance of the proposed method remains very competitive.

One possible drawback of the use of a mixture of Gaussians could be the potentially larger number of parameters to be estimated, contrary to the proposed sparse representations of ISRFs in a dictionary that use only a small number of atoms (typically around four). Nevertheless, we believe that a study of the use of mixtures of Gaussians in this context could be an interesting future research direction. This perspective was included in the conclusion of the paper in page 21.
* * *
[a]If the referee has any reference on this approach applied to the ISRF estimation, we would be grateful for the opportunity to conduct a comparative analysis with our proposed approach.

**Comment 1.5**

The authors said that the analysis of concentrations from two spectrometers could provide a better understanding of the carbon cycle. Nevertheless, it seems that the analysis in the paper has been done separately (ie., per instrument). Is it possible to use data fusion in this case in order to have a more accurate results?

**Response:**

We thank the reviewer for this remark. The associated sentence was misleading, and we corrected it into: "Note that OCO-2 and MicroCarb aim to provide a better understanding of the carbon cycle". The two spectrometers are used for the same purpose. However their design is different and the associated ISRFs to be estimated are different. Thus, the analyses are conducted separately for these two spectrometers.

**Comment 1.6**

The authors should add a supplementary material, or annex in order to elaborate more on the algorithm, specifically, in page 6, it is not clear how the matrix S is constructed, how we compute the "appropriate" error matrix ... A pseudo code would be much appreciated from the readers

**Response:**

Following the reviewer's comment, we have added a pseudo-code for the K-SVD algorithm in Appendix A5 (page 22-24). The appendix also describes the computation of the associated error matrix, based on the algorithm defined in [1].

**Comment 1.7**

In page 7, the authors want to assess the robustness of the proposed ISRF. What do you mean exactly by robustness (is it w.r.t. the noise level, a possible mismatch, a possible presence of outliers ...), can you be more specific?

**Response:**

The use of the word "robustness" in page 7 was indeed misleading. There, we are not assessing the "robustness" of the proposed method w.r.t. noise levels, mismatches, etc., but we are demonstrating the applicability of the method to different spectrometers. The word "robustness" was thus modified to "applicability".
* * *
**Comment 1.8**

The caption of some figures is too short and needs more explanation, eg, Fig 1, Fig. 2, Fig 9
Typo in eq 5 (=)

**Response:**

We thank the referee for this comment. The typo was corrected, and the captions of the figures have been modified in order to include more explanations as:

- "Examples of MicroCarb ISRFs." becomes ". Illustration of a superposition of 1024 ISRFs with centered wavelengths $\lambda_l = 758.3,...,768.3$nm around their central wavelengths.The ISRFs have been simulated for the band B1 of the MicroCarb instrument using uniform scenes." (page 5)

- "Representation of the four first atoms of the dictionary constructed using one SVD (top) or using the K-SVD algorithm (bottom) for the MicroCarb spectrometer (band B1)." becomes "Representation of the four first atoms of the dictionary of ISRFs $\mathbf{\Phi}$ constructed using an SVD on the matrix of representative ISRFs (top) or using the K-SVD algorithm using the same matrix of representative ISRFs (bottom) for the MicroCarb spectrometer (band B1)." (page 7)

- "Examples of ISRFs IN, scene and NU (MicroCarb band 1)." becomes "Examples of ISRFs from uniform scenes (ISRF IN - left) and from different non-uniform scenes displayed in Fig. 11 and FOVs (ISRF scene - right) (MicroCarb band B1)."

**2 Response to referee RC2 ♯2**

**Comment 2.1**

The authors present a novel method to estimate the Instrument Spectral Response Function (ISRF) of pushbroom-like spectrometers using a data driven approach. They propose to model the ISRF as sparse linear combination of an over-complete set (dictionary) of basis functions (atoms) derived from laboratory characterization measurements. The proposed method is applied to estimate the ISRF of one ground-based (Avantes) and 5 space-borne imaging spectrometers (OMI, TROPOMI, GOME-2, OCO-2 and MicroCarb) designed for remote sensing of the Earth's atmospheric composition. The results obtained for these spectrometers are compared to a state-of-the-art reference ISRF model (fit of a generalized Gaussian). The proposed new algorithm outperforms the reference method in all cases presented by the authors.

Accurate post-launch ISRF estimation is a prerequisite for the delivery of accurate atmospheric remote sensing products and the proposed approach is a novel and creative contribution which should be of interest to the community. The presented investigation seems thorough and the main ideas are presented clearly. Despite some minor issues in some equations, the mathematical basis is outlined sufficiently well. The results shown generally support the authors' claims.

**Response:**

We thank the reviewer for his/her appreciation of our work.

**Comment 2.2**

There are two major issues, which need to be addressed in my opinion:
Firstly, the authors claim (quite correctly), that a continuous post-launch monitoring of the ISRF is required, because instruments change over time (e.g. due to thermal breathing) and because the ISRF is usually scene dependent, because an inhomogeneous along-track illumination of the spectrometer entrance slit leads to a different effective ISRF than the one typically measured during on-ground pre-flight characterization with a homogeneous along track illumination. However, according to my understanding, the training data set used to compile the dictionary of atoms suggested by the authors consists exclusively of laboratory measurements obtained under homogeneous illumination conditions in the laboratory (except for a very limited study for the MicroCarb instrument in chapter 5.4.3). As such, I would expect these measurements to neither contain effects caused by thermal drift nor those caused by inhomogeneous along track illumination (within one pixel). As the usefulness of the proposed method greatly depends on its behavior under these typical conditions, I would recommend to add results obtained with synthetic data (e.g. by changing the FWHM of the ISRFs of the simulated spectra) to analyze the performance of the proposed algorithm under real-world conditions.

**Response:**

We thank the reviewer for this comment. It is true that thermal variations have not been simulated for the data considered in this work. However, it was observed for the MicroCarb mission that the shape of the ISRFs is not very dependent on thermal variations (test at 160K, 165K). Conversely, ISRF variations due to luminance (spectral axis - along track) or according to the observed scene are much more significant and were considered in our work. The proposed methods could thus be applied in the presence of thermal variations.

The effect of inhomogeneous scenes along the track illumination on the MicroCarb instrument is considered in Sect. 5.3.3 of the manuscript. Note that several ISRFs on the spectral axis should be affected by this error, not just one pixel. If, as suggested, an error is made within a pixel, it is assumed to be an error in the instrument's detector, and the main assumption that the ISRF does not change much in a small observation window no longer holds (the variation of ISRF shapes in the window is too far from 1%). A more detailed study of this observation has been included in Sect. 5.3.4 of the manuscript, where a different ISRF has been generated for a given pixel and the robustness to pixel errors is assessed. This ISRF is presented in Fig. 14 (page 20) of the manuscript. As shown in Fig. 15 (page 20), the introduction of an erroneous ISRF within the sliding window results in an increase of the estimation error. This observation indicates that it is no longer feasible to estimate the erroneous ISRF with a low error. Note that the estimation error and the associated residuals still provide insight into the underlying issue, which is an inherent inaccuracy in the instrument's detector. This sparse representation-based approach can thus also be used to detect instrument-related errors. All in all, the conclusion of this work is that, as long as the observation window assumption is valid and the dictionary is sufficiently diverse, methods based on sparse representation outperform those based on parametric models.

**Comment 2.3**

Secondly, I have difficulties understanding, how exactly the training data set was obtained from the laboratory measurements and how the spectra used for validation were generated. Simply stating that the ISRFs were "obtained by spline interpolation" is not sufficient in my opinion without explaining in which dimension and at which nodes the interpolation was carried out.

**Response:**

We thank the reviewer for this comment and have clarified this point as follows. For each instrument, the associated ISRFs were recovered with the corresponding wavelengths. However, the number of ISRFs retrieved, after removing the ISRFs associated with bad pixels, is less than the number of pixels. Therefore, to identify the ISRF at the missing nominal wavelengths $\lambda_l$, a linear interpolation of ISRFs at two specified nominal wavelengths was considered, as recommended in the OMI slit function product (https://www.knmiprojects.nl/projects/ozone-monitoring-instrument/data-products/omslit). The ISRF at $\lambda_l$ is defined by

$$I_l = \frac{\lambda_l - \lambda_a}{\lambda_b - \lambda_a} I_b + \frac{\lambda_b - \lambda_l}{\lambda_b - \lambda_a} I_a.$$

The number $N_\lambda$ of wavelengths may differ from the number of pixels of the instrument, a situation that occurs when the ISRF associated with the first and/or last pixels are not available. This information on the generation of the data has been included in Sect. 5.2.2 of the revision, when presenting the OCO-2 instrument and in the Supplement.

**Comment 2.4**

Additionally, I am not sure, whether the ISRFs chosen for the validation were part of the training data (modulo noise) or whether additional changes were introduced (e.g. those mentioned above, leading to shapes and FWHMs typically not encountered during on-ground CAL). As I would expect the behavior of the proposed algorithm to depend strongly on the choice of training data (see chapter 5.4.3), this is a critical issue which has to be addressed before publication in my opinion as one strength of the reference (Gaussian) model is its independence of prior knowledge in this regard. Consequently, a fair comparison should investigate these scenarios, which are of high practical importance.

**Response:**

As pointed out by the reviewer, the behavior of the developed method indeed depends on the choice of the training method. In the case of scenes that exhibit minimal variation in ISRFs (e.g., desert scenes), a dictionary constructed from ISRFs associated with these uniform scenes can be employed to identify them. To construct this dictionary, only 10% of the total number of ISRFs are used. The process becomes more intricate when the scenes are no longer uniform since the ISRF shapes are more diverse, including asymmetric variations, as illustrated in Sect. 5.3.3. However, our results demonstrate that incorporating just three additional ISRFs associated with these non-uniform scenes leads to accurate estimations.

Finally, the statement that the Gaussian and Super-Gaussian models do not require any assumptions regarding the shape of the ISRFs is somewhat misleading to the authors' opinion. Indeed, when using these parametric models, there is a strong prior assumption on the shape of the ISRF (given by the definition of these parametric models), which is, as demonstrated in the article, not always correct. Furthermore, the parametric models yield inferior results in the uniform scenarios, which suggests that their performance in non-uniform scene scenarios may also be limited. The proposed method makes it possible to remove the assumption of Gaussian shape for the ISRFs and to adapt to the shape of the ISRFs in a non-parametric way.

**Comment 2.5**

Additionally, it is my impression, that the authors have a very comprehensive knowledge of the Micro-Carb mission, which clearly emerges when discussing results related to this specific instrument. Maybe it is worth considering whether the suggested publication could be dedicated entirely to MicroCarb? Demonstrating that the proposed ISRF retrieval method works under a broad variety of conditions for this instrument alone would convince me of its value. I feel, that the studies presented for the other instruments add little substance to an already convincing demonstration in this regard. This might also help to shorten the manuscript and allow to add currently incomplete or ambiguous information.

**Response:**

Our work proposes a new methodology for estimating ISRFs that can be applied to various instruments. Yet, as suggested by the reviewer, in order to better balance the length and focus of the manuscript, the proposed revision mainly concentrates on the MicroCarb instrument, and we have included only the additional results obtained for a single additional instrument (OCO-2) in a new section 5.2.2 (page 12). Since the potential applicability of the proposed method to other instruments (and other designs) is also a significant contribution and a key aspect of this work, we have moved the results related to the instruments Avantes, GOME-2, OMI and TROPOMI into a separate supplementary material.

**Comment 2.6**

2/4: The optical layout described here is basically a pushbroom spectrometer. In principle, other designs are in use as well (e.g. FTIR), which have to be treated differently. Maybe clarify to which spectrometer types your method applies.

**Response:**

It is true that the spectrometers used in this study are passive pushbroom spectrometers (mainly hyperspectral dispersive spectrometers). Indeed alternative designs, such as FTIR, are also employed in practice and are presented in a footnote of Sect. 4.1 (Instruments, datasets & preprocessing) of the paper. These instruments employ a Michelson interferometer, and the ISRFs can be obtained through the inverse Fourier transform. However, this inversion is applied when the optical path difference (OPD) is assumed to be constant, which is not always the case since the OPD may vary depending on the position (rendering the use of the Fourier transform to find the ISRF no longer applicable). Moreover, in certain applications, undersampling may be necessary. In such cases, applying the Fourier transform becomes more challenging. The issue can then be modeled using a linear inverse model, such as the one considered in this study, and the proposed ISRF estimation process can be used. In summary, the proposed method is not limited to specific types of instruments and does not require many details about the instrument. A contribution of this work is that it can be applied to any instrument as long as the problem can be formulated as an inverse problem and the following hypotheses hold:

1. A sufficient number of measurements associated with the same ISRF are available (either because the ISRFs do not vary much in a small observation window, in the spectral or spatial domain, or because observations from several reference spectra can be obtained for the same ISRF).

2. A sufficient amount and variety of ISRFs have been identified and characterized on the ground to construct the dictionary.

**Comment 2.7**

2/4: Additionally, the telescope creating the virtual image in the slit does not necessarily image the spectrally dispersed scene on the detector (MicroCarb is rather an exception than the rule in this regard). Maybe add a sketch showing the essential principle design you are investigating?

**Response:**

As suggested by the reviewer, the design of the MicroCarb instrument, obtained from [2], was added to the paper and is displayed in Fig. 3 in Sect. 4.2 (page 9) of the revised manuscript.

**Comment 2.8**

4/1-4: What exactly do you consider similar w.r.t. the ISRF and how do you formalize this mathematically? The following unnumbered equation seems to suggest that neighboring ISRFs are assumed to be equal (not just similar) inside a window of Nobs bands. Within an accuracy of 1% I would challenge this assumption for the instruments under consideration.

**Response:**

We thank the reviewer for this comment and have clarified this point in the section 2.1 (page 4) of manuscript. The proposed model is indeed applied to ISRFs, which exhibit slight variations along the spectral axis. It is expected that the average of the normalized absolute error between the ISRFs in a window of $N_{\text{obs}} + 1$ observations and the central ISRF at wavelength $\lambda_l$ is below a specified threshold for the ISRF estimation error, i.e. 1%. Note that the larger this variation, the more important the discrepancies in ISRF shapes. The small variation assumption is not valid for the whole set of wavelengths and the size of the sliding window must be adjusted in order to solve the ISRF estimation problem. Alternatively, observations from $N_{\text{obs}}$ reference spectra could be used when available.

**Comment 2.9**

Additionally, could you simply solve this problem (a constant ISRF for multiple bands) using Fourier Transform / Wiener Filter without further assumption on the shape of the ISRF ?

**Response:**

Indeed, for some spectrometers (e.g., FTIR), it is possible to use the Fourier Transform or the Wiener Filter to retrieve the ISRFs. However, in the present case, this is not possible because each observation is associated with a single point from a convolution with different portions of the spectrum. Moreover, the number of observations is smaller than the number of unknown ISRF points ($N > N_{\text{obs}}$). The methodology for addressing this type of problem is to either assume a parametric model (e.g., Gaussian or Super-Gaussian models) or a non-parametric model with a sufficiently small number of unknown parameters to be estimated. The proposed approach solves this problem by making use of sparse representations in a dictionary, which yield competitive results with respect to Gaussian and super-Gaussian models. This observation about FTIR spectrometer was added in the footnote 3, page 8 of the revised manuscript.

**Comment 2.10**

Why do you need one equation per band (l) instead of a single equation / matrix including all bands simultaneously ?

**Response:**

The OMP algorithm inherently requires independent ISRF estimations, thus leads to solving $N_\lambda$ independent inverse problems (one problem per band). In principle, the problem could also be formulated using a single equation, for all bands jointly. However, this would lead to a significantly more complicated problem and estimation algorithm, which is left for future work. We will outline possible research directions related to this perspective in the manuscript, stating that a prospect for research could be to estimate all the ISRFs simultaneously by introducing or learning a new regularization term to account for the ISRF variation along the wavelengths.

**Comment 2.11**

4/24-26: It is not obvious to me, why an ISRF model of two (generalized) Gaussians with slightly shifted center wavelengths would be insufficient to model the displayed ISRF. Please elaborate.

**Response:**

The study of an ISRF model using two generalized Gaussians has not been considered before to our knowledge and is an interesting suggestion. Based on the reviewer's comment, we have included new results into Section 5.2.1 of the revised manuscript: As displayed in Fig. 6 in page 12 of the manuscript, this novel parametric approach yields enhanced outcomes as compared to the classic Gaussian and Super-Gaussian methods. However, the performance of these models is not competitive with respect to sparse representation based methods. One potential avenue for further investigation would be the use of a mixture of Gaussians and generalized Gaussians for the construction of the dictionary. However, this represents an alternative formulation of the problem that has not yet been explored, and to the best of our knowledge, has not been previously investigated. This perspective was included in the conclusion of the paper.

**Comment 2.12**

4/24-26: Are the shown ISRFs obtained under homogeneous illumination or are they part of the "ISRF Scene" examples shown in fig. 9? I would suggest to specify the band / channel / geom. pixel combination for all shown ISRFs to eliminate ambiguity.

**Response:**

We thank the reviewer and we agree that more details about the ISRF presented in Fig. 9 were required. This figure illustrates how the ISRFs can vary depending on the variety of scene types and illuminations. The left ISRFs are representative of uniform scenes, whereas the right ISRFs correspond to non-uniform scenes (the slit was not uniformly illuminated during the integration time). The caption of the figure 11 in the revised manuscript (page 18) was modified to read "Examples of ISRFs from uniform scenes (ISRF IN - left) and from different non-uniform scenes displayed in Fig. 12 and FOVs (ISRF scene - right) (MicroCarb band B1).".

**Comment 2.13**

Entire Chapter 4: I think more details regarding the generation of the reference spectra is required here:

Which parameters are chosen for the radiative transfer simulations (trace gas concentration profiles, aerosols, scattering, surface albedo, ...)? Which SNR was assumed to generate the noise? Does the SNR change with sensor and/or wavelength? (ref. section 5.4.1) Except from adding noise, are you only using ISRFs included in the training data set (dictionary) or are you also creating reference spectra with ISRFs slightly narrower or wider than the training data (e.g. to simulate sharpening or blurring caused by thermal breathing of the instrument)? If so, how are those modeled ? Do the simulations include the effect of (along-track) surface albedo inhomogeneity within one pixel ? If so, on which length scale / sampling distance ?

**Response:**

We thank the reviewer for this remark. The description of the methodology employed in the generation of the reference spectra has been detailed and clarified in Section 4.1 of the revised manuscript (page 8): The profiles originate from the Thermodynamical Initial Guess Retrieval (TIGR) database, which is hosted by Aeris data (https://www.aeris-data.fr/en/projects/thermodynamical-initial-guess-retrieval-tigr/). A sample from the database is selected for the generation of the reference spectrum. The measured spectra are obtained by convolving the reference spectrum with the ISRFs. In order to assess the impact of noise on the measured spectra, a series of Gaussian noise simulations were conducted, the results of which are presented in Section 5.3.1. of the revision. In practice, the method is applied to measured spectra from which the noise has already been reduced (using spatial binning, preliminary estimation).

**Comment 2.14**

How exactly (along which dimension and at which sampling points) do you interpolate the ISRFs for each instrument? Why does $N_\lambda$ differ from the number of spectral bands for some instruments?

**Response:**

For each instrument, some pixels are referred to as bad pixels or have high errors. The associated ISRFs are thus discarded. Then, as explained in Comment 4, the ISRFs are interpolated between two given nominal wavelengths $\lambda_a$ and $\lambda_b$ for all wavelengths $\lambda_l$ using the following interpolation formula $I_l = \frac{\lambda_l - \lambda_a}{\lambda_b - \lambda_a} I_b + \frac{\lambda_b - \lambda_l}{\lambda_b - \lambda_a} I_a$. For some instruments, $N_\lambda$ differs from the number of spectral bands because no extrapolation was made to retrieve the ISRFs outside the nominal wavelengths. This observation was explained in the Section 5.2.2 (page 12) and in the supplementary material.

**Comment 2.15**

I think the description of each instrument in a separate sub-section does not add a lot of relevant information beyond what can be found in the cited literature. Have you considered summarizing the relevant information (number of spectral bands, source for the ISRFs, reference citation for the instrument/mission) in a table (and remove sections 4.2 to 4.7)? I think this might enhance clarity and readability of the manuscript.

**Response:**

As suggested, only the OCO-2 instrument is studied in addition to the MicroCarb instrument in the revision (see also our answer to Comment 2.5). The results obtained for the other instruments have been included in a supplementary material. A table summarizing the information on the different instruments is provided to enhance clarity and readability.

**Comment 2.16**

10/second equation: Why are you including the entire sliding window into the error measure? Would it not be sufficient and more meaningful to compute the difference between measured spectrum s and simulated spectrum r $\hat{I}$ at the center wavelengths $\lambda_l$ of each channel l? The Nobs nodes left and right of $\lambda_l$ are only used as computational aid as far as I understand and may e.g. increasingly suffer from boundary effects when approaching the limits of the window. Why do you include these effects into the error measure? In order to support a direct comparison with the assumed SNR, I think a relative measure would be desirable as well (ref table 1).

**Response:**

We thank the reviewer and agree that the information presented in the results section (Sect. 5.2) was not sufficiently explained. The second figure simultaneously shows the absolute differences between the normalized spectral measurements and their normalized approximations (in logarithmic form, second rows) and the corresponding mean residual, represented by the symbol $\rho$. To enhance clarity, for each spectrometer, the proposed figures now show the measured spectrum reconstruction, the difference between the measured spectrum and the estimated one, the residuals for each wavelength, which allows the reader to visualize the function which is minimized, the ISRF approximation error and the mean ISRF approximation error versus the number of selected atoms as displayed in Fig. 5 (page 11) and 8 (page 14) of the revised manuscript. In this study, the ISRF estimation process is conducted through the minimization of an objective function which is defined using the $N_{\text{obs}}$ observations. Displaying the results of the minimization process is important in our opinion for comparison with the outcomes of the alternative methods.

**Comment 2.17**

12/fig(3e): How is it possible, that the Gaussian (as special case of the super-Gaussian) fits the data that much better than the super-Gaussian here? Are you sure the fit converged properly?

**Response:**

It is indeed not expected that the Gaussian distribution outperforms the Super-Gaussian. The initial insight for answering this question is that, with regard to this particular spectrometer, the ISRFs (illustrated in Fig. 4 of the revised manuscript) appear to be more Gaussian than the ISRFs of the other spectrometers, allowing a better convergence for the Gaussian model. The second insight is that the estimation problem that is solved, using either the Gaussian or the Super-Gaussian parameterisation, is formulated as a nonlinear least-square problem. To solve it, we employ a simplex-based optimization method (MATLAB function *fminsearch*) to minimize the residuals between the measured and estimated spectra using the parametric models. The super-Gaussian method is initialized using the results obtained with the Gaussian model. This method usually provides better performance than the Gaussian model. However, the simplex-based optimization method does not always converge to the optimal solution, which is the case in this example. This observation was added in Sect. 5.2.2 (page 13) of the revised manuscript.

**Comment 2.18**

How are the ISRFs in this figure normalized (not unit area) ?

**Response:**

In all results, the ISRFs have been normalized to unit area.

**Comment 2.19**

Chapter 5.2 and 5.3.x: Lacking the information listed above, I feel I cannot comment on the authors' claims here in a meaningful way, as it is of fundamental importance how the reference spectra were chosen and whether they are included in the training data or not.

**Response:**

The authors hope that the information provided on the reference spectra in previous comments (Comment 13, which profiles are obtained using the TIGR database) will help the reader to better understand and appreciate our claims. However, one significant contribution of this work is that as long as the reference spectrum is known, it is possible to estimate the ISRFs with high accuracy. The reference spectrum is supposed well-known (obtained from well-known scenes such as desert, moon, solar scenes) and this spectrum is not included in the training (only ISRFs characterized on the ground are). It is in principle possible to use any known reference spectra with sufficient local variations in its spectral content.

**Comment 2.20**

Chapter 5.4.1:
Does SNR=50 dB imply SNR=100 000? (which seems quite high to me). In this case I would expect a log10 residual for a "perfect fit" around -5 if sufficiently many atoms are chosen, but the values in fig. 6 (b) are significantly higher. Could you elaborate on this? Looking at table 1, it seems to me that the SPIRIT approach works significantly less effective for SNR = 100 (20 dB), which is not uncommon for many earth observing instruments. Could you comment on the usability of SPIRIT in these scenarios? Chapter 5.4.3: Why do you choose across-track binning into these exact pixel groups? Are the ISRFs for the bands in these geometric regions similar?

**Response:**

The noise considered in our experiments has been generated for different SNR levels. We agree that there is no discernible difference in the results obtained with SNRs of 80dB and 120 dB. Consequently, all results obtained with SNR=120dB have been removed in the revision. In practice, a spatial binning is typically employed to enhance the signal-to-noise ratio (SNR). This binning involves the arbitrary division of the imaged area on Earth into distinct field of views (FOVs) (3 for instance for MicroCarb). The measured spectrum for each FOV is obtained as an average of the measured spectra within that FOV. This is achieved through the application of the following formula: $s(\lambda_l) = \frac{1}{C} \sum_{c=1}^{C} s_c(\lambda_l)$, where $s_c(\lambda_l)$ is the value of the measured spectrum associated with the wavelength $\lambda_l$ at the spatial pixel $c$ in the FOV. It is indeed assumed in this study that the ISRFs do not change much in across-track (spatial axis) as compared to along track (spectral axis). In the case of the MicroCarb mission, the binning represents a compromise between the objective of achieving a satisfactory signal-to-noise ratio (SNR) and that of maintaining a suitable ground grid, which has a resolution of 13.5 km in ACT and 9 km along the track. These comments about the SNR and the binning have been included in Sect. 5.3.1 of the revision, where the robustness to noise is evaluated.

**Comment 2.21**

How exactly are the ISRFs for inhomogeneous scenes obtained? Any simulation would require knowledge of the ISRFs for an inhomogeneously illuminated pixel/slit I presume. Is this knowledge inferred from measurements or simulations based on an optical instrument model?

**Response:**

For a given reference spectrum, non-uniform scenes are generated using asymmetric ISRFs. In practice, there is no information available regarding the non-uniformity of a given scene from the measured spectra. It is only during the inversion process, when estimating the ISRFs, that it becomes apparent that these ISRFs have been modified. More details can be found in [3]. This observation about the generation of ISRFs for non-uniform scenes has in a footnote in Sect. 5.3.3 (page 16) of the revision.

**Comment 2.22**

1/2-3: optical elements "induce errors in the measurement"? Then why not leave them out ;-)? Maybe rephrase?

**Response:**

We thank the reviewer for pointing us to this ambiguous phrase. The sentence "Spectrometers are composed of different optical elements that can induce errors in the measurements and therefore need to be modeled as accurately as possible" was modified by "Spectrometers are composed of different optical elements and detectors that must be modeled as accurately as possible." (page 1 of the revision).

**Comment 2.23**

2/9: Is an ISRF not rather associated with a channel than a wavelength? The (center) wavelength of said channel can then be chosen based on (mean, max or median of) the ISRF?

**Response:**

Indeed, the ISRF is associated with a given pixel, designated by the index "$l$". The wavelength associated with the pixel is then obtained as the center (maximum, median, or barycenter) of the ISRF at the given pixel. However, there are some effects, such as the smile (in ACT) or some gaps in our knowledge about the wavelengths (in along track), that can result in spectral shifts, which can degrade the estimation of ISRFs. The reason behind stating that each pixel is associated with a single ISRF and wavelength is that the influence of these various factors is still under investigation and not accounted for in this work. Consequently, we assume that the wavelength is known and address the ISRF estimation problem by solving an inverse problem. This observation was incorporated in Sect. 2.1 during the ISRF estimation problem as a footnote (page 3) and the potential impact of spectral shifts was outlined as a future objective of the research.

**Comment 2.24**

2/14: ... "ISRF wavelength variations exceed this threshold" ? Which wavelength varies here? How does this affect the ISRF error budget ?

**Response:**

This sentence was in fact misleading. The sentence "For some missions, ISRFs are expected to be known with a normalized error less than 1%, which is a challenge since the ISRF wavelength variations exceed this threshold" was modified to "For some missions, ISRFs are expected to be known with a normalized error less than 1%, which represents a significant challenge given that the variations in ISRF shape across the entire band frequently exceed this threshold."

**Comment 2.25**

2/17: I would argue, that all pushbroom instruments are susceptible to effective ISRF changes if the illumination varies within a pixel (along-track), as partial illumination of the spectrometer entrance slit is equivalent to a narrower slit and thus (usually) a smaller FWHM. Unless optically mitigated (e.g. by means of a slit homogenizer or optical fibers), this effect has to be taken into account in the error budget. Does the 1%-requirement include the associated uncertainties?

**Response:**

It is evident that the ISRF's dependence on the scene can impact a multitude of instruments. This represents a design choice for the instruments. It might have been feasible, for example, for the MicroCarb mission to defocus the instrument in order to avoid this dependence on the slit illumination. However, the introduction of a defocus can potentially compromise the precision of the instrument, and thus it was ultimately decided to exclude this option. The performance of 1% on the ISRF knowledge is a consequence of the MicroCarb mission's necessity for an accurate determination of $CO_2$ concentrations. This performance is an objective of the mission, but it can indeed be challenging to achieve in practice using real data. The 1% requirement accounts for uncertainty, acquisition noise of ISRFs and interpolation. In this study, however, the 1% requirement is a defined threshold for the maximum of the ISRF approximation error. This observation was incorporated in Sect. 5.3.3, when the robustness to ISRF changes is assessed as a footnote (8 in page 16).

**Comment 2.26**

3/9: Technically each channel / geometric pixel combination has an individual ISRF. This ISRF can then be used to define a center wavelength for each channel of each geometric pixel. Have you considered associating each ISRF with the number (l) of a channel instead of the center wavelength? This might simplify the notation in many equations in my opinion.

**Response:**

As discussed in Comment 2.24, we could define the ISRF using the number ($l$) of the channel instead of the associated wavelength. However, we wanted to model the problem at first sight by being as close as possible to the physical phenomenon behind (in Eq. (1) : $s(\lambda_l) = (r * I_l)(\lambda_l)$), which implies the use of a wavelength. Afterward, the ISRF is defined as a vector using the number $l$ only.

**Comment 2.27**

3/eq(1): The convolution is usually defined over the entire space of real numbers. Also, the ISRF I is mirrored along the wavelength axes in this notation, as I(u) is the sensitivity to the wavelength $\lambda_l$ - u, which might be unexpected for many readers. It has no practical effect on your results of course.

**Response:**

Based on the reviewer comment, we have changed the notation of eq(1) in Sect. 2.1 accordingly.

**Comment 2.28**

3/eq(2): I think a $\Delta$ is missing in front of the sum. Also: Does $\lambda_l - n\Delta$ equal $\lambda_{l-n}$ ? If not: How do you choose $\Delta$?

**Response:**

It is possible that a misunderstanding has occurred. It is assumed that the ISRFs are defined on a regular grid of wavelengths centered at 0. The associated wavelength gap between two ISRF measurements is designated as $\Delta$. The expression $\lambda_l - n\Delta$ is thus not equal to $\lambda_{l-n}$ given that the space between two points in the ISRF is smaller than that between two points in the measured spectrum. The sentence following Eq. (2) in Sect. 2.1 (page 3) "where $\Delta$ is the wavelength sampling interval for the ISRFs, which is assumed to be regularly sampled." has been modified in the revision to: "where $\Delta$ is the sampling period between two consecutive points of the ISRF, which is assumed to be regularly sampled.".

**Comment 2.29**

3/21: "A major difficulty with the inverse problem ...": Maybe also mention another very fundamental problem: Eq. 1 is a Fredholm equation, the solution of which is the classical example of an ill-posed problem. Additionally the constraint of a single measurement could be removed experimentally using e.g. the sun and a spectrally tunable on-board calibration source, albeit at extra cost.

**Response:**

Thank you for your comment regarding the Fredholm equation. However, we have chosen not to address this issue in the revision, as our study is based on a single observation obtained from the convolution product. The focus of the article is on formulating the problem as a linear inverse problem to be estimated, rather than solving the Fredholm equation. The information on the constraint of a single measurement which could be removed experimentally using e.g. the sun and a spectrally tunable on-board calibration source, albeit at extra cost, has been added in Sect. 2.1 as a footnote (n°2 in page 3).

**Comment 2.30**

4/eqs(3 & 4): Why $x \in \Delta$ (i.e. the sampling interval) ? Could you not use any real number for x? (For most atmospheric spectrometers the sampling ratio is greater 2, so the FWHM exceeds one sampling interval in most cases.)

**Response:**

It is true that $\Delta$ is the sampling interval. However, $\boldsymbol{\Delta}$ also represents the wavelength grid, i.e., the points where the ISRFs are defined. The equation: "$\boldsymbol{\Delta} = [-\frac{N}{2}\Delta, ..., \frac{N}{2}\Delta]$" was not correct and has been replaced by: "$\boldsymbol{\Delta} = \{-\frac{N}{2}\Delta, ..., \frac{N}{2}\Delta\}$" in Sect. 2.1 (page 3) of the revision.

**Comment 2.31**

4/eq(5): I think there is one superfluous equation symbol following the summation symbol and the summation index n should probably occur somewhere in the equation (unless the summation is part of the higher dimensional 2-norm)?

**Response:**

Corrected as suggested.

**Comment 2.32**

10/5: "carbon" without trailing "e"

**Response:**

Corrected as suggested.

**Comment 2.33**

10/first equation: If the ISRFs are normalized to unit area, can the denominator have values different from one?

**Response:**

We thank the reviewer for the remark, this equation is the usual expression used to assess the performance of the ISRF estimation method. However, the ISRFs are indeed normalized to unit area. The denominator has been removed from the revision and the following sentence has been added next to the equation: "For the selected instrument, the ISRFs are assumed to be normalized to unit area." in Sect. 5.1 (page 9) of the revision.

**Comment 2.34**

12/fig(3): The font is barely legible and my aging eyes can hardly discriminate the two LASSO variants. Please increase the size of these plots. The Avantes SVD/K-SVD fits seem to have linear segments, which I would not expect in a "real" ISRF. Are these linear interpolation artifacts? Maybe add markers at the computational nodes for clarity?

**Response:**

The size of the plot has been increased and some markers have been added for the Avantes SVD/K-SVD fits.

**Comment 2.35**

13/fig(4): Even at a zoom level of 190 % I can barely read this figure! Please increase the size of this figure and the font size.

**Response:**

We apologize for the lack of readability. The size of the figure has been increased accordingly.

**Comment 2.36**

20/fig(9): Please indicate for which band / channel / pixel combination these ISRFs are valid.

**Response:**

This figure (fig. 11 in page 18 of the revised manuscript) is composed of two distinct sets of ISRFs. The first set, displayed in the left figure, comprises ISRFs from uniform scenes, randomly selected from the total set of 1024 ISRFs. The second set displayed in the right figure comprises ISRFs from non-uniform scenes, randomly selected from the total set of eight scenes and three FOVs. The figure caption has been modified from "Examples of ISRFs IN, scene and NU (MicroCarb band 1)" to "Examples of ISRFs from uniform scenes (ISRF IN - left) and from different non-uniform scenes displayed in Fig. 12 and FOVs (ISRF scene - right) (MicroCarb band B1)."

**Comment 2.37**

20/fig(10): Do the figures on the left-hand side show a single pixel IFOV in along-track / across-track direction? Otherwise please indicate the pixel limits. How do these images enter into the ISRFs shown on the right? Are these ISRFs simulated or measured ?

**Response:**

The image displayed in this figure (fig. 12 in page 19 of the revised manuscript) shows a scene that is directly observed by the spectrometer's slit and subsequently recorded by the instrument's detector during the integration period. This image is then obtained in the ACT direction. From this image, a cut is made along the ACT direction into three equal parts, resulting in three FOVs. The resulting ISRFs have been simulated.

**Comment 2.38**

17/table 1: I think it would be helpful to indicate the expected minimum error for a given SNR. Considering an ideal Gaussian with synthetic noise at an assumed SNR of 1000 I would e.g. expect a mean residual error of approx. 0.1 %, but I do not observe a relation of this kind in the data. Could you elaborate on the relationship between SNR and normalized approximation error a little bit more in the text?

**Response:**

As suggested by the reviewer, the anticipated minimum error for a specified SNR has been identified in the revision. However, regardless of the method employed, the ISRFs are estimated by using a model. Thus, it is inevitable that their estimation will not be perfect, even in the absence of noise. In the table, the term "normalized approximation error" refers to the discrepancy between the approximated and the actual ISRFs. In order to avoid any confusion, "Normalized approximation error," has been changed to "Mean ISRF approximation error."

**Comment 2.39**

18/fig(7) & 19/fig(8): Considering the limited range of the error values, I would recommend a linear scale for the y-axis.

**Response:**

Modified as suggested.

**Comment 2.40**

21/fig(11): This figure is also quite small. Maybe also consider a linear scale. Is it necessary to resolve differences smaller than the noise level?

**Response:**

The size of the figure has been increased. However, it was decided to keep the scale in log10 as for the other results in order to see the differences in the ISRF approximation errors.

**3 Response to Community Comment CC1 ♯3**

**Comment 3.1**

The use of an iterative and dictionary-based based approach for estimating ISRFs is a rather original solution.

**Response:**

We thank the community member for his/her positive appreciation of our work.

**Comment 3.2**

The fact that the most simple method, SVD + OMP, eventually leads to the best performance is a very good, yet somehow surprising, news, and could be further commented: has this to do with a particular choice of the hyper-parameter in (8) or with a lack of discrimination of the L1 norm constraint?

**Response:**

There is, indeed, no theoretical reason for the method SVD + OMP to lead to superior performance when compared to working with the $\ell_1$ norm penalty. In our study, the hyperparameters for the algorithms have been chosen in order to achieve the best results. However, the OMP and LASSO algorithms address two distinct problems. Indeed, when the OMP algorithm provides an approximate solution to the problem with the $\ell_0$ penalty, the LASSO algorithm solves the problem using the $\ell_1$ penalty, hence gives the solution of an alternative, different problem. Certains limitations of the LASSO algorithm have been highlighted in numerous publications, including [4], and may be at the origin of our observations. This observation about the comparison in performance between OMP and LASSO was further explained in conclusion of the results in Sect. 5.2.3 (page 14) of the revision.

> **Comment 3.3**
>
> The iterations between dictionary estimates and sparse approximation represent an important aspect of the study, and could be further described.
>
> > **Response:**
> >
> > Thank you for pointing this out. The following pseudo-codes have been added as an Appendix A (page 22-25) in the revision of the manuscript:
> >
> > - A1. The generation of the matrix of reference spectrum.
> > - A2. The construction of the dictionary.
> > - A3. Orthogonal Matching Pursuit (OMP) algorithm.
> > - A4. LASSO algorithm (for which the MATLAB *lasso* function is used).
> > - A5. Construction of the dictionary using the K-SVD algorithm. (based on the algorithm defined in [1]).

> **Comment 3.4**
>
> The adaptation to calibration errors, and temporal drifts of the feature represent high potential perspectives for this work.
>
> > **Response:**
> >
> > Thank you for this important suggestion. We are indeed planning to study how our method could be adapted to calibrations errors, and temporal drifts, and will elaborate on this in the Conclusions and Perspectives of the article.

**References**

[1] M. Aharon, M. Elad, and A. Bruckstein, "K-SVD: An algorithm for designing overcomplete dictionaries for sparse representation," *IEEE Trans. Signal Process.*, vol. 54, no. 11, pp. 4311–4322, 2006.

[2] M. Castelnau, E. Cansot, C. Buil, V. Pascal, V. Crombez, S. Lopez, L. Georges, and M. Dubreuil, "Modelization and validation of the diffraction effects in the microcarb instrument for accurately computing the instrumental spectral response function," in *International Conference on Space Optics—ICSO 2018*. SPIE, 2019, vol. 11180, pp. 1054–1068.

[3] C. Pittet, V. Crombez, D. Jouglet, L. Georges, E. Cansot, and A. Albert-Aguilar, "In-flight estimation of the microcarb instrument spectral response functions.," in *Geophysical Research Abstracts*, 2019, vol. 21.

[4] R. Tibshirani, "Regression shrinkage and selection via the lasso," *Journal of the Royal Statistical Society (Series B)*, vol. 58, pp. 267–288, 1996.